# 4D analysis of malaria parasite invasion offers insights into erythrocyte membrane remodeling and parasitophorous vacuole formation

Niall D. Geoghegan[1,2,5], Cindy Evelyn [1,2,5], Lachlan W. Whitehead [1,2], Michal Pasternak [1,2,3], Phoebe McDonald[1,2], Tony Triglia[1,2], Danushka S. Marapana[1,2], Daryan Kempe[4], Jennifer K. Thompson[1], Michael J. Mlodzianoski [1,2], Julie Healer[1,2], Maté Biro [4], Alan F. Cowman [1,2] & Kelly L. Rogers [1,2 ✉]

Host membrane remodeling is indispensable for viruses, bacteria, and parasites, to subvert the membrane barrier and obtain entry into cells. The malaria parasite *Plasmodium spp.* induces biophysical and molecular changes to the erythrocyte membrane through the ordered secretion of its apical organelles. To understand this process and address the debate regarding how the parasitophorous vacuole membrane (PVM) is formed, we developed an approach using lattice light-sheet microscopy, which enables the parasite interaction with the host cell membrane to be tracked and characterized during invasion. Our results show that the PVM is predominantly formed from the erythrocyte membrane, which undergoes biophysical changes as it is remodeled across all stages of invasion, from pre-invasion through to PVM sealing. This approach enables a functional interrogation of parasite-derived lipids and proteins in PVM biogenesis and echinocytosis during *Plasmodium falciparum* invasion and promises to yield mechanistic insights regarding how this is more generally orchestrated by other intracellular pathogens.

[1] The Walter & Eliza Hall Institute of Medical Research, Parkville, VIC, Australia. [2] Department of Medical Biology, The University of Melbourne, Parkville, VIC, Australia. [3] Imperial College London, London SW7 2AZ, UK. [4] EMBL Australia, Single Molecule Science Node, School of Medical Sciences, University of New South Wales, Sydney, NSW, Australia. [5] These authors contributed equally: Niall D. Geoghegan, Cindy Evelyn. ✉email: rogers@wehi.edu.au

Malaria is caused by parasites of the genus *Plasmodium* with *P. falciparum* causing >400,000 deaths globally each year[1]. In humans, symptoms of the disease manifest when the asexual blood-stage merozoite invades and multiplies within erythrocytes[2]. Micron-sized merozoites are released into the bloodstream from *P. falciparum*-infected erythrocytes, and their surface proteins enable attachment to nearby erythrocytes[3,4]. The merozoite induces rapid and localized membrane deformations on the erythrocyte surface, enabling reorientation of its apical end towards the host membrane[5,6]. Reorientation is believed to be driven by a concentration gradient of adhesins secreted by the microneme and rhoptry organelles onto the merozoite surface, such that the highest concentrations are at the apical end[7]. A variety of ligand–receptor interactions involving the adhesins are also responsible for conditioning the host cell for invasion. As an example, the micronemal protein EBA175 binds glycophorin A on the erythrocyte surface[8], inducing biophysical changes to the host membrane and downstream phosphorylation of cytoskeletal proteins[9,10].

Difficulties in the development of anti-malarial therapies or vaccines persist, in part, due to high polymorphisms among parasite ligands and redundancy of some ligand–receptor interactions at these early stages of invasion[11]. PfRh5, a rhoptry-derived protein, is an essential invasion ligand of *P. falciparum*[12–14]. PfRh5 binds the basigin receptor on erythrocytes, and this is linked to a calcium ($Ca^{2+}$) flux at the parasite-host membrane junction[15,16]. Thereafter, the tight junction is formed via the interaction of PfAMA1 and the RON complex[17,18] and it is here where the parasite anchors its actin–myosin motor, providing the driving force for membrane invagination[19,20]. At the completion of internalization, the invaginated membrane seals behind the parasite forming the parasitophorous vacuole[21].

The host erythrocyte, by its very nature, is non-endocytic and thus the internalization of the micron-sized merozoite requires significant perturbation and morphological upheaval of the underlying membrane and cytoskeleton. Ultrastructural images captured by electron microscopy[22,23], and more recently by super-resolution immunofluorescence microscopy[19], show lipid and proteins are secreted by the parasite rhoptries onto the erythrocyte during invagination. Evidence for rhoptry lipid secretion, appearing as multilamellar membranous whorls, has given rise to the suggestion they are a major element of the newly formed parasitophorous vacuole membrane (PVM)[24,25]. However, later studies on both *Plasmodium spp.* and *Toxoplasma gondii* suggested the PVM is derived from host cell membrane[26,27]. The most-prevailing hypothesis is a biophysical model of membrane-wrapping describing the energetics of invasion with emphasis on energy contributions from parasite-induced host cell remodeling, which includes host cytoskeletal disruption and parasite membrane secretion onto the host membrane[7]. Cholesterol enrichment and the presence of host detergent-resistant membrane-associated proteins in the nascent PVM[28] are indications of membrane remodeling and its role in PVM formation. Indeed, the depletion of cholesterol from the erythrocyte membrane has an inhibitory effect on parasite invasion and growth[29], but how it is recruited to the PVM and at which step it is involved in invasion remain unclear[30].

The challenge of studying PVM formation lies in the dynamic nature of the event, which is completed in 10–20 s[31], and the lack of an established model for membrane invagination. Here, using lattice light-sheet microscopy (LLSM), we developed a method to capture the process of invasion using fast volumetric imaging. This gentle microscopy approach allowed all stages of invasion to be quantitatively characterized, including the kinetics of parasite internalization and PVM formation. Our results show that the nascent PVM is predominantly formed from the erythrocyte membrane and that it is continuously remodeled throughout invasion. Indicative of this is the enrichment of cholesterol in the newly forming PVM and observations that RON3, a rhoptry bulb protein, is released onto the invaginated membrane during internalization. Our studies support the view that the tight junction acts both as an anchor for the parasite to drive in and a molecular boundary to ensure a targeted and dynamic remodeling of the invaginated membrane[32]. An interplay between erythrocyte membrane remodeling and the force generated by the parasite actin–myosin motor must therefore occur for the completion of parasite entry.

## Results

**Quantitative analysis of *P. falciparum* invasion in four dimensions.** Most time-lapse recordings of *Plasmodium* invasion of erythrocytes are undertaken using differential interference contrast (DIC) or widefield microscopy, where the newly formed PVM is largely inaccessible and volumetric information is unavailable[15]. To overcome these limitations, we custom-built a dual-camera LLSM system, as previously described[33], to image *P. falciparum* invasion of erythrocytes in four-dimensions (4D). To surmount the dynamic nature of this process, occurring over tens of seconds to minutes, and the small size of the parasite, LLSM offered a high enough level of resolution to visualize invasion in detail. Compared to methods we had previously optimized for invasion experiments, such as resonant scanning confocal microscopy, LLSM provided volumetric data with a high spatiotemporal resolution, a significantly larger field of view, a better signal-to-noise ratio, and less photobleaching (Supplementary Figure 1). Furthermore, a distinct advantage of this 4D data was that it could be viewed in multiple orientations relative to the invading parasite due to the near isotropic resolution (Fig. 1a, Supplementary Movie 1, and Supplementary Figure 1b).

We labeled the erythrocyte membrane with the long-chain carbocyanine dye, PKH26, or with the potentiometric steryl dye, Di-4-ANEPPDHQ. Invasion of an erythrocyte by a merozoite could be captured with subcellular resolution across all stages, commencing with egress (Supplementary Movie 1) and followed by merozoite–erythrocyte interactions, which produced transient deformations and sometimes marked ridges or folds on the surface of the erythrocyte membrane (Fig. 1a–e). A kinetic analysis of the forming PVM, allowed quantitative comparison of invasion kinetics over multiple events (Fig. 1b, c, $n = 12$). The plots showed transient peaks during the pre-invasion stage corresponding to membrane deformations (Fig. 1c(i)) and demonstrated a high degree of uniformity during the linear phase of internalization (Fig. 1c(ii)). Internalization was observed to be complete at the point at which the curve reaches a plateau (Fig. 1c(iii)), indicating a fully formed PVM. Variations in the amplitude and width of the peaks during the pre-invasion phase, which correspond to the strength and duration of membrane deformations, had no correlation to the rate of invasion (Fig. 1d, e).

We were also able to track the parasite's motion within the newly formed PVM. Mitotracker dyes gave an asymmetric labeling pattern, which enabled the dynamics of three key stages to be characterized: (i) host cell recognition and reorientation, (ii) internalization, and (iii) intra-PVM dynamics. Interestingly, we noted that parasites became dynamically active within the parasitophorous vacuole and exhibited a twisting motion upon the completion of the internalization phase, as demonstrated by a series of oscillations on the curve (Fig. 1f, g and Supplementary Movie 2). Similar behavior has also been described for *Toxoplasma gondii*[34] and this is thought to be a mechanical

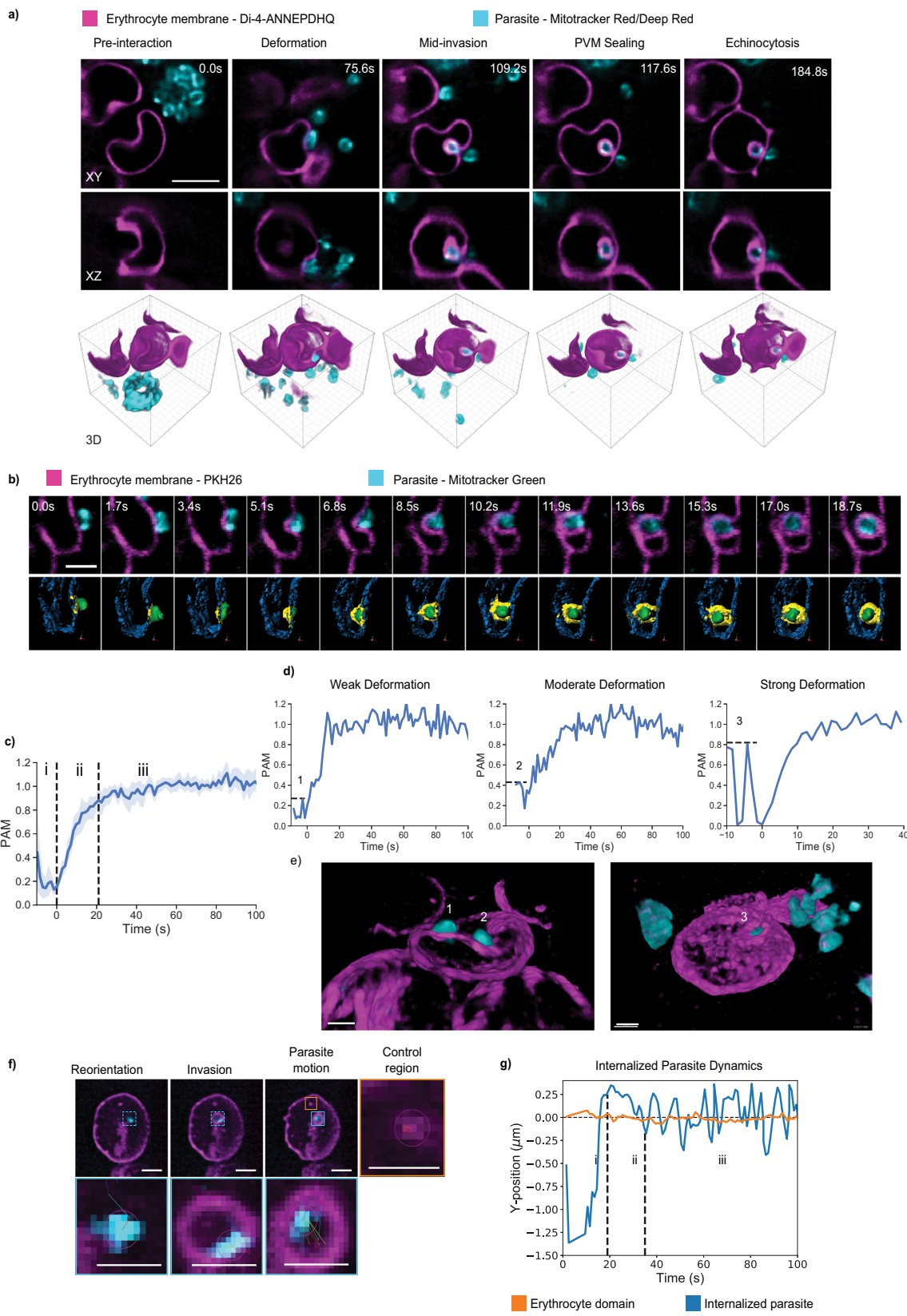

mechanism for the parasitophorous vacuole to seal and eventually undergo scission from the host cell membrane. LLSM, therefore, afforded the possibility to quantify all stages of invasion in 4D and to visually interrogate PVM biogenesis biophysically and mechanistically.

**The nascent PVM is predominantly formed by the host erythrocyte membrane.** An outstanding question in the field of erythrocyte invasion by *P. falciparum* is the origin of the PVM. Some studies suggest that the PVM is formed largely through invagination and scission of the host membrane[26,27], however,

**Fig. 1 LLSM provides quantifiable imaging of *P. falciparum* invasion of erythrocytes in 4D with subcellular detail. a** Representative snapshots from LLSM showing the key stages of invasion in XY, XZ, and 3D-sectioned views, including pre-egress, deformation, mid-invasion, PVM sealing, and echinocytosis. Images were acquired at a rate of one volume per 8.4 s with 168 planes spaced 158 nm ($n = 9$ independent experiments). Scale bar: 6 µm. **b** Representative time-lapse images showing segmentation of host membrane, cyan, in contact with the invading parasite, green, delineates the formation of the PVM providing quantification of the parasite-associated host membrane (PAM), yellow, with 1.7 s time interval. Scale bar: 2 µm. **c** Normalized increase in PAM over time for $n = 12$ invasions. Values represent mean ±95% confidence interval (CI), demonstrating a conserved dynamic of PVM formation. (i) shows pre-invasion deformations, (ii) shows internalization process, and (iii) shows post-internalization period. **d** Normalized PAM for three individual invasion events from **c** preceded with (1) weak, (2) moderate, and (3) strong deformations, **e** with corresponding 3D views. Scale bar: 2 µm. **f** Representative maximum intensity projection images showing tracking of an invading parasite and one region of high-intensity host lipid domain, scale bars: 2 µm, **g** with the corresponding plot for Y-position. Blue plot shows the Y-position of the parasite during (i) pre-invasion and reorientation, (ii) internalization phase, and (iii) intra-PVM dynamics. The orange plot shows Y-position of the host lipid domain over time for comparison. All images displayed and analyzed using IMARIS. **b–g** Representatives from five independent experiments.

others postulate that parasite-derived lipid is the dominant contributor to PVM biogenesis[24,25]. To address this question, 4D LLSM provided data demarking the relative surfaces of the host erythrocyte and PVM in high resolution through erythrocyte membrane labeling. Cellular segmentation was performed using the LimeSeg[35] plugin in ImageJ/FIJI[36] allowing for reliable membrane boundary determination (Fig. 2a). This novel segmentation, coupled with the near isotropic Nyquist sampled 3D LLSM data, enabled measurements of surface area and volume of the invaded host erythrocyte and newly formed PVM (Fig. 2b). To validate this approach, we began by performing measurements on healthy erythrocytes over a broad field of view to determine the distribution of surface area values and found that they were similar to previous measurements (150 µm², $n = 10$)[37]. Applying this approach to invasion showed that invaded erythrocytes demonstrated a stepwise reduction in surface area, including those for which there was a single or double merozoite invasion event on a single erythrocyte (Fig. 2c and Supplementary Movie 3). We quantified the loss of membrane surface area on 16 erythrocytes and showed that the reduction in erythrocyte surface area is related to the total sum of surface area calculated for the PVM of each parasite (Fig. 2d). These data suggest that the PVM is predominantly composed of host erythrocyte material. Reduced volume represents the volume ratio between the host erythrocyte and a perfect sphere with the same surface area. In all cases, a rise in the reduced volume was seen coinciding with the surface area loss from PVM formation, suggesting that the cell begins undergoing the transformation to an echinocyte immediately following the invasion, as the infected cell conforms to a more spherical morphology (Fig. 2c). This event, therefore, occurs much earlier than previously thought[15]. These two quantifiable parameters, surface area, and reduced volume, provide a simple geometric description of key steps in the invasion process, which have been traditionally difficult to assess. These data imply that the parasite-derived lipids likely play a more functional role during the invasion process. We, therefore, hypothesize that the parasite-derived lipid material is important for further remodeling of the nascent PVM during invasion and downstream of vacuole sealing.

**A Ca²⁺ flux at the merozoite apex sheds light on tight junction formation.** Previous studies report a punctate $Ca^{2+}$ flux preceding penetration of the parasite into the host erythrocyte[15,16]. The precise origin and function of this event are unknown, largely owing to its transient and localized nature. However, the general view is that this event follows the essential PfRh5–basigin interaction and marks the formation of a fusion pore between the host erythrocyte and the rhoptry organelles of the invading parasite[15,16]. It should therefore be an excellent indicator for assessing the inhibition of invasion downstream of this event when combined with LLSM. Although some reports have observed the $Ca^{2+}$ flux

event in only 45% of cases, these studies used single plane widefield microscopy and consequently only a small depth of field[15,38]. To address this limitation, we used LLSM to image parasites invading erythrocytes loaded with Fluo-4AM to capture whole volumes of $80 \times 50 \times 10$ µm (140 slices per frame) and observed the flux in 89% of cases (Fig. 3a, $n = 28$). Our 4D data also show that the $Ca^{2+}$ flux is localized to the merozoite apical end on the extracellular side of the erythrocyte membrane at the parasite-host interface (Supplementary Figure 2a). This provides further evidence for pore formation between the parasite apex and the erythrocyte membrane that enables the passage of Fluo-4AM from the host cytosol to the merozoite apical organelle, from which $Ca^{2+}$ is then being released into the host cell[15,16].

Despite the tradeoff in temporal resolution (one volume every 2.8 s), we were still able to identify all stages of invasion, including host cell deformations, merozoite apical reorientation, PVM formation, and echinocytosis. For all invasion events, the timing of the $Ca^{2+}$ flux coincided with a final, and often intense, deformation of the membrane preceding the commencement of internalization (Supplementary Figure 2b). We termed this the recoil phase, as in most cases the parasite was observed to pull back on the membrane, with only its apical end firmly attached (Supplementary Movie 4). This is analogous to previous data where we showed a backward pinch on the erythrocyte membrane at the attachment stage using stimulated emission depletion microscopy, where RON4 is docked, ready for insertion[16], and also to electron microscopy data shown by others where RON4 is underneath the erythrocyte membrane[19]. Taken together, the spatiotemporal specificity of the punctate $Ca^{2+}$ flux described here suggests that the apical attachment via PfRh5–basigin interaction is achieved during a final deformation, which triggers a pore formation on the parasite-host interface. Upon the erythrocyte membrane recoil, the parasite releases $Ca^{2+}$ from its apical organelle, coinciding with the expected timing for tight junction formation before it propels itself into the host cell.

We, therefore, used the $Ca^{2+}$ flux as a temporal marker to assess changes in the invasion kinetics downstream of the PfRh5–basigin interaction in the presence of known inhibitors. We imaged invasion in the presence of R1 peptide, which blocks tight junction formation, and cytochalasin D, which inhibits actin polymerization of the parasite motor. In agreement with previous reports[15], merozoites were still able to attach to a nearby erythrocyte and induce a $Ca^{2+}$ flux in the presence of these inhibitors, however, they were unable to invade (Fig. 3b, c, Supplementary Figure 3b, c). Plotting the kinetics of invasion over time relative to the $Ca^{2+}$ flux highlighted the effects of R1 peptide and cytochalasin D inhibition on invasion (Fig. 3d). As expected, deformations were less apparent when the actin–myosin motor was unable to engage in the presence of cytochalasin D. However, in the presence of R1 peptide, deformations occurred with varying intensities after the $Ca^{2+}$ flux, indicating the parasite's attempt for

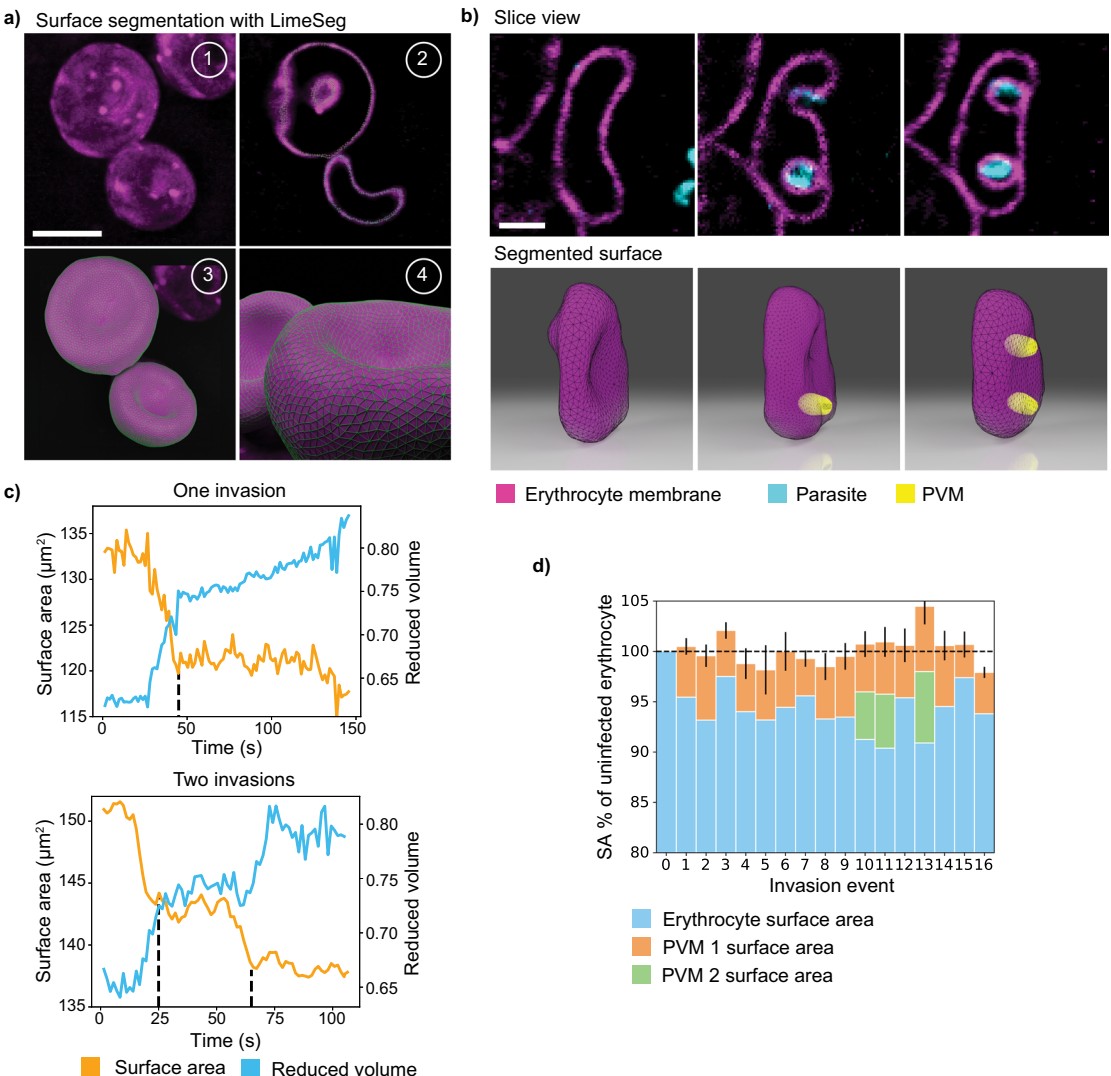

**Fig. 2 The PVM is predominantly composed of host cell material as revealed by surface segmentation and analysis. a** Representative images showing the process of surface segmentation on erythrocytes using LimeSeg[36]. Panels represent the (1) maximum intensity projection (MIP), (2) mid-slice, and (3–4) segmented surface of two uninfected erythrocytes. Scale bar: 5 μm. **b** An erythrocyte invaded by two parasites. XY mid-slices showing pre-invasion, first invasion, and second invasion of the erythrocyte followed by the segmented surfaces of the host erythrocyte and the two distinct PVMs. MIP and mid-slice images were displayed using ImageJ/FIJI and 3D views were created with Blender. Scale bar: 2 μm. **c** Temporal tracking of the host erythrocyte membrane surface area and reduced volume during invasion. Examples of invasion by one and two parasites are displayed, with the dashed line marking the time where the parasite is fully internalized. **d** Surface area measurements of infected erythrocytes and their respective PVMs, each normalized to the surface area of the erythrocyte before invasion ($n = 16$). Error corresponds to the standard deviation (SD) of the mean erythrocyte surface area, taken from five pre-invasion frames, added with the SD of the cumulative surface area of the erythrocyte and the PVMs, taken from five post-invasion frames. The analysis shown here was performed on data from four independent experiments.

internalization (Supplementary Movie 5). Using a custom-written MATLAB-based algorithm, we then generated surface maps of the local Gaussian curvature of the host cell to compare the local deformations induced by the invading parasite with and without R1 peptide. A tightly constricted region of negative Gaussian curvature is present for the untreated parasites as the tight junction maintains the nexus through which the parasite invades. In contrast, strong deformations were induced by the R1 peptide-treated parasites, but the negative Gaussian curvature around the deformation rim was absent (Fig. 3e and Supplementary Movie 6). These data show that the force of the parasite motor and strength of adhesion from high-affinity receptor-ligand interactions alone, is not enough to drive the wrapping of the erythrocyte membrane around the parasite surface. This supports the view

that the AMA1–RON2 interaction is needed to tightly link the erythrocyte membrane to the surface of the parasite and to anchor the parasite motor during internalization[7,32]. In addition, the high level of negative Gaussian curvature would also suggest that the AMA1–RON2 interaction is coupled to the disruption of the underlying cytoskeleton, as this would facilitate the degree of membrane invagination needed for internalization. It remains to be determined what role the AMA1–RON2 interaction has in the reorganization of the host cell cytoskeleton at the site of invasion. The use of $Ca^{2+}$ flux as a temporal reference, combined with the quantitative analyses enabled by LLSM, could provide the modality to explore the effects of other inhibitors or knockout lines on the kinetics of invasion, which would add to our insights on the invasion mechanism.

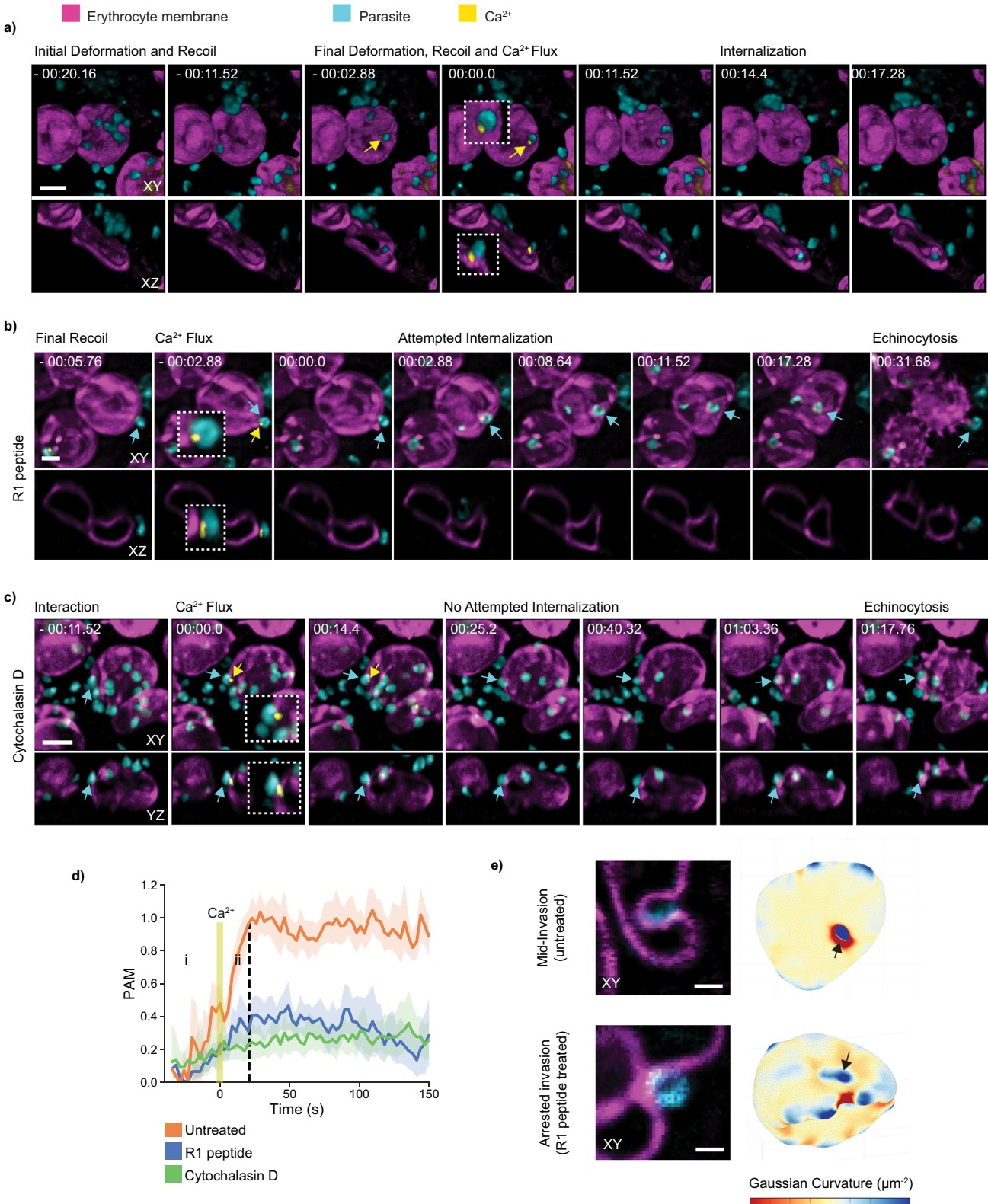

**Membrane remodeling has an important role in echinocytosis.** A poorly understood stage of the invasion process is that of echinocytosis. Echinocytosis is transient in nature, lasting for a few minutes before the invaded cell returns close to its original morphology[6]. Studies have previously suggested a role for Ca[2+] in the process, but are yet to find a clear link[15]. One possible hypothesis has been that activation of the Gardos channel,

responsible for potassium (K[+]) efflux and water loss, induces echinocytosis. However, we have found no link between inhibition of the Gardos channel and the erythrocyte's ability to undergo echinocytosis (Supplementary Figure 4). Building on our findings, particularly regarding the observed change in reduced volume relative to the surface area, we next investigated whether echinocytosis was directly linked to membrane remodeling

**Fig. 3 The punctate Ca$^{2+}$ flux marks the formation of a fusion pore during host membrane recoil and acts as a temporal reference for analysis of downstream inhibitors.** Images represent parasites labeled with MitoTracker Red, cyan, interacting with erythrocytes labeled with Di-4-ANEPPDHQ, magenta, and loaded with Fluo-4AM Ca$^{2+}$ reporter, yellow, displayed using IMARIS in extended XY view and XZ view. **a** Sequence of invasion steps including the punctate Ca$^{2+}$ flux localized at the parasite apical tip ($n = 25$). Parasite-induced deformations lead to a final erythrocyte membrane recoil that coincides with the Ca$^{2+}$ flux prior to internalization. Time-lapse of parasite-host interaction **b** in the presence of R1 peptide ($n = 22$), which blocks tight junction formation, and **c** cytochalasin D ($n = 17$), which disrupts actin polymerization on the parasite. Parasites were still able to trigger Ca$^{2+}$ flux and induce echinocytosis despite failing to internalize. **a–c** Scale bars: 2 μm. **d** Parasite-associated host membrane (PAM) time plot demonstrating a conserved timing of the Ca$^{2+}$ flux coinciding with host membrane recoil, marking the transition from deformation (i) to internalization (ii) ($n = 12$). For the R1 peptide ($n = 9$) and cytochalasin D-treated ($n = 6$) events, plots were shifted in time accordingly with the Ca$^{2+}$ flux being the reference point. Line with shaded error band represent mean ±95% CI. PAM values were normalized to a fully internalized parasite from the untreated data sets. **e** Representative XY slices and 3D images of Gaussian curvature mapping on erythrocyte with red-blue look-up table from the untreated and R1 treated parasite–erythrocyte interaction reported in **a**, **b**. The untreated parasite induced a highly negative curvature around the aperture of the invagination pit, which was absent from the deformation induced by the R1 peptide-treated parasite. Scale bars: 1 μm.

during invagination and formation of the PVM. We pre-treated the erythrocytes with methyl-β cyclodextrin (MβCD) to alter their membrane composition and we used either R1 peptide or cytochalasin D to inhibit invasion. Using the punctate Ca$^{2+}$ flux as a temporal marker, we marked the timing of the point at which undulations on the erythrocyte membrane were first observed (echinocyte I) and the point at which the erythrocytes had transformed into a spherical cell with 10 or more spicules (echinocyte III) for each condition[39]. In all cases, and independent of PVM formation, echinocytosis occurred with a varying degree of delay after the Ca$^{2+}$ flux (Fig. 4a). We also noted that the length of time taken for the erythrocyte to transform from echinocyte I to echinocyte III varied across the treatments (Fig. 4b). A significant difference in the time from the initiation of Ca$^{2+}$ flux to the beginning and duration of echinocytosis was seen for both MβCD and cytochalasin D treatment. The transformation to echinocytosis occurred more rapidly with MβCD treatment. In contrast, cytochalasin D inhibition of invasion significantly prolonged the process of echinocytosis.

Another notable observation was the presence of membrane protrusions in both R1 peptide- and cytochalasin D-treated samples (Supplementary Movie 7). In the case of R1 peptide, long membrane tethers on the exterior part of the host cell were observed to be protruding from the host membrane, adjacent to the parasite attachment site, and sometimes wrapping the parasite (Fig. 4c). In contrast, cytochalasin D treatment induced similar structures, but they projected from the parasite into the interior of the host cell (Fig. 4d). The formation of these tethers and protrusions coincided with echinocytosis, indicating a localized disruption to the host membrane bilayer. This suggests that the release of lipid-rich rhoptry bulb-derived material, known to coincide with tight junction formation[19], initiates the transformation of the erythrocyte from its typical biconcave shape to an echinocyte. This is consistent with a previous study showing a knockout of the rhoptry bulb protein, RAMA, causes mislocalization of RON3 and prevents echinocytosis[40].

To determine the timing and the role of rhoptry bulb release in PVM remodeling, we used CRISPR-cas9 to generate a parasite line with a mNeonGreen-tagged RON3 protein. As expected, RON3 was found to adopt a rhoptry bulb membrane localization[41] in the mature schizont stage and within free merozoites (Fig. 4e). We found that during invasion RON3 was redistributed to the PVM as the parasite invaginated the host cell membrane (Fig. 4f and Supplementary Movie 8). A PVM localization for RON3 is consistent with its significant effect on glucose uptake across the PVM, as shown by a recent study using a RON3 knockout parasite line[42]. In the presence of R1 peptide, we observed a tether, extending extracellularly, down which RON3 was transported and continued to be distributed to the host membrane until the erythrocyte underwent echinocytosis (Fig. 4g). This demonstrates

that rhoptry bulb release occurs independent of tight junction formation, however, if the release occurs after tight junction formation, the parasite-derived proteins would be confined to the growing vacuole membrane. We made similar observations by performing immunofluorescence assays on fixed infected red blood cell preparations. These data provide additional evidence that the mNeonGreen-tagged RON3 protein is redistributed to the PVM upon invasion (Supplementary Figure 5). The data resemble what has been seen with RAP1 in fixed samples[19]. Like RON3, RAP1 is also a rhoptry bulb protein, which is released into the host cell. However, in the presence of R1 peptide, it was also shown to be aberrantly released onto the erythrocyte surface[19]. LLSM together with a fluorescently tagged rhoptry bulb protein reporter line, such as the mNeonGreen-RON3 line described here, could enable this mechanism to be interrogated in three dimensions and promises to address some of the outstanding questions regarding echinocytosis.

Together with the surface area and reduced volume data, our results, therefore, suggest that echinocytosis is triggered by parasite-induced membrane remodeling, either involving the ejection of rhoptry bulb components into the erythrocyte membrane, which follows the PfRh5–basigin interaction[15], or the formation of PVM at the completion of internalization. This is demonstrated under normal conditions and also when invasion is blocked downstream from the PfRh5–basigin interaction, such as in the presence of cytochalasin D and R1 peptide. Our observations that echinocytosis occurs more rapidly in MβCD treated cells may be relevant to the perturbed membrane composition, which could render cells more sensitive if additional lipid is then integrated into the membrane bilayer. On the other hand, echinocytosis is both delayed and prolonged when cells are treated with cytochalasin D, which could be linked to the release of rhoptry bulb material into the erythrocyte membrane, after and beyond the tight junction.

**The nascent PVM is dynamically remodeled from early stages of invasion.** Although our data strongly suggest that the PVM is largely formed from host membrane material, there is evidence in the literature that the nascent PVM is rich in detergent-resistant membrane-associated proteins and cholesterol[28,29]. This could be linked to the high membrane curvatures produced during invasion, which might lead to the mechanical redistribution of host membrane lipids and their associated proteins or it may be due to the active transport of these lipid domains into the newly forming PVM. This suggests there is early remodeling of the PVM during the invasion process. Our observation that there is localized membrane disruption in the presence of invasion inhibitors and that echinocytosis may be a lipid-driven process suggests that the remodeling process begins early after the commencement of internalization. To address this question, we measured local

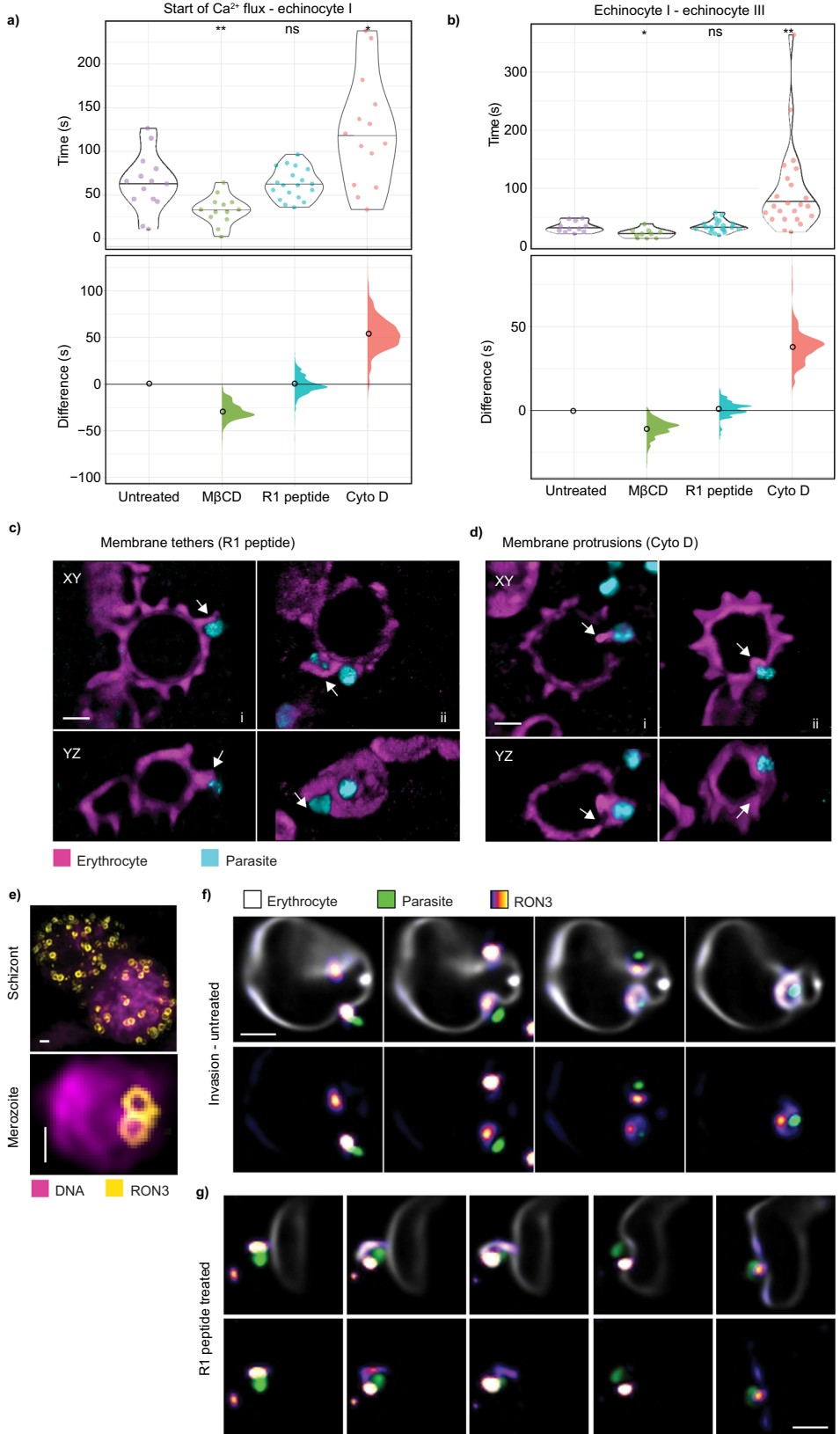

changes in the membrane composition of the forming PVM during invasion. To temporally measure changes in lipid composition, we employed a technique based on the ratiometric imaging of a functional fluorescent dye, in this case, Di-4-ANEPPDHQ. The dye has previously been used to probe lipid order in model and plasma membranes[43,44]. Solvatochromic dyes, such as Di-4-ANEPPDHQ, are often used as a proxy for lipid order by reporting the relative hydration level of the dye molecules in the bilayer. Recent investigations have, however, found that the response of Di-4-ANEPPDHQ is more specific to cholesterol itself rather than a measure of lipid order when compared with the well-characterized probe Laurdan[45]. We validated this by

**Fig. 4 Echinocytosis is likely induced by membrane fusion and lipid redistribution from parasite to host cell during rhoptry release. a** Time differences between the start of $Ca^{2+}$ flux to host cell transformation into echinocyte I across different treatments. Untreated ($n = 14$), MβCD ($n = 13$, $p = 0.005$), R1 peptide ($n = 19$, $p = 0.618$), cytochalasin D ($n = 14$, $p = 0.04$). **b** Time differences for the transformation from echinocyte I to echinocyte III across different treatments. Untreated ($n = 11$), MβCD ($n = 10$, $p = 0.032$), R1 peptide ($n = 20$, $p = 0.762$), cytochalasin D ($n = 23$, $p = 0.01$). *$p < 0.05$; **$p < 0.01$; two-sided randomization test. **a**, **b** The central lines and empty circles represent mean values. **c**, **d** Extended XY and YZ views, showing merozoite–erythrocyte interaction in the presence of inhibitors. R1 peptide treatment results in a membrane tether between the parasite and erythrocyte following echinocytosis, observed in three independent experiments. Cytochalasin D treatment results in observable membrane thickening forming intracellular protrusions from the attempted invasion site, observed in two independent experiments. Data were displayed in blend mode and processed using IMARIS. Scale bars: 2 μm. **e** Representative images from two independent experiments showing RON3 localization on late-stage schizont and free merozoite, captured with structured illumination microscopy. Scale bars: 0.5 μm. **f**, **g** Snapshots showing the release of RON3, a rhoptry bulb protein, by untreated and R1 treated parasites. Images captured with LLSM, denoised using content-aware image restoration (CARE)[62], and displayed as slice view with FIJI. In the control event, RON3 was released onto the invagination pit and incorporated into the parasitophorous vacuole membrane as the parasite invades. With R1 inhibition, RON3 was released onto the erythrocyte membrane instead. This observation was followed up with an immunofluorescence assay in fixed cells (see Supplementary Figure 5). Scale bars: 2 μm.

altering the erythrocyte membrane cholesterol, using MβCD, and lipid order, using sphingomyelinase, before labeling the erythrocytes with Di-4-ANEPPDHQ. The generalized polarization (*GP*) value of the labeled erythrocytes was measured and used as an indicator for the lipid composition of treated and untreated cells. As expected, the *GP* value significantly decreased when cholesterol is removed and remained constant when lipid order is disrupted (Supplementary Figure 6). This suggests that the primary mechanism for changes to the dye's *GP* value comes from some interaction with cholesterol itself.

Upon validation of the dye as a cholesterol-sensitive probe, we sought to measure dynamic remodeling of the host cell membrane during the steps of invasion. This was achieved by adaptation of the segmentation method used in Fig. 1. Cholesterol distribution in Di-4-ANEPPDHQ-labeled erythrocytes was quantified during merozoite invasion. Using this approach, we demonstrate that there is a dynamic reorganization of the host membrane and developing PVM during invasion (Fig. 5a). At the outset of internalization, an increase in *GP* value of 0.05 arbitrary units was measured from the membrane region in contact with the invading parasite. It was seen that the average *GP* value within this segmented region continued to increase relative to the rest of the host erythrocyte membrane throughout the entire invasion process and for some time following the sealing of the PVM (Fig. 5b and Supplementary Movie 9). These data present a clear view of dynamic membrane remodeling during invasion and that the nascent PVM environment is rich in cholesterol. Our results support previous hypotheses that cholesterol-enriched microdomains from the host membrane are recruited to the vacuolar membrane together with detergent-resistant membrane-associated proteins, as the parasite invades[28–30,46].

Cholesterol depletion of erythrocytes using MβCD showed a dose-dependent reduction in parasite growth (Fig. 5c). We, therefore, performed live-imaging invasion assays on erythrocytes pre-treated with MβCD at a concentration that was found to cause ~50% growth inhibition, using the $Ca^{2+}$ flux as a marker for parasite commitment to invasion. The invasion assays showed a reduced percentage of parasites inducing $Ca^{2+}$ flux that leads to the successful invasion on the cholesterol-depleted erythrocytes. The unsuccessful invasion events include cases where the parasite either failed to internalize or was ejected several minutes after the parasite had become fully internalized (Fig. 5d and Supplementary Movie 10), suggesting a lack of sustained enrichment in cholesterol may destabilize PVM formation. Cholesterol depletion of erythrocytes has been shown to reduce invasion in another study, however, the step at which the invasion is halted was not determined[29]. In our studies, 4D microscopy suggests that the redistribution of erythrocyte membrane cholesterol has a role in conditioning the erythrocyte membrane at different stages of

invasion, particularly during internalization. In particular, the reorganization of the erythrocyte membrane through the enrichment of cholesterol would help to drive higher membrane curvatures in the host membrane. Finally, the redistribution of cholesterol and associated proteins may be crucial for the final step that leads to membrane fusion and sealing of the PVM. Overall, the ability to probe the complex and dynamic process of invasion in such exquisite detail allows us to add new insights to the existing model of *P. falciparum* invasion of erythrocytes (Fig. 6).

## Discussion

The application of high spatiotemporal resolution LLSM allows previously inaccessible elements of *P. falciparum* invasion into erythrocytes to be quantitatively analyzed. Through segmentation and tracking of the forming PVM, we can determine both the degree of deformation and the rate of invasion. These key metrics are often assessed in the invasion-based assay when measuring the effect of therapeutics or knockout strains of *P. falciparum*, but largely in a qualitative manner. The near isotropic nature of the data also allowed the relative surface area and volume of the invaded erythrocyte to be quantified over time. Altogether, this provides the strongest evidence to date that the PVM is constitutively composed of host cell material, which settles a decades-long debate as to the dominant contributor of the PVM[32,47]. What remains is the question regarding the role of parasite lipids and proteins that are ejected into the erythrocyte membrane during invagination[22].

LLSM reports on the previously described $Ca^{2+}$ flux[15] and shows that it is localized to the parasite apical end, at the external side of the erythrocyte membrane. This supports a previous hypothesis that a pore is established, permitting the passage of Fluo-4AM from erythrocytes into the neck of the $Ca^{2+}$ rich rhoptry compartment[15,16]. Further studies will need to determine whether the rhoptry organelle is indeed a bona fide $Ca^{2+}$ store. We also made the unexpected observation that the flux coincided with membrane recoil and relaxing of the host membrane to its native curvature before the commencement of internalization. The release of $Ca^{2+}$ across the host cell membrane overlaps with the known timing of tight junction formation and rhoptry bulb release. We expect that some form of disruption to the underlying erythrocyte cytoskeleton would be needed to support the high degree of curvature achieved during membrane invagination. Either coupled to tight junction formation, or acting independently, it remains to be determined if the $Ca^{2+}$ flux of material ejected from the rhoptry organelle is responsible for disrupting the underlying cytoskeleton at this stage of invasion. Nevertheless, the $Ca^{2+}$ flux serves as an excellent marker for the kinetics of

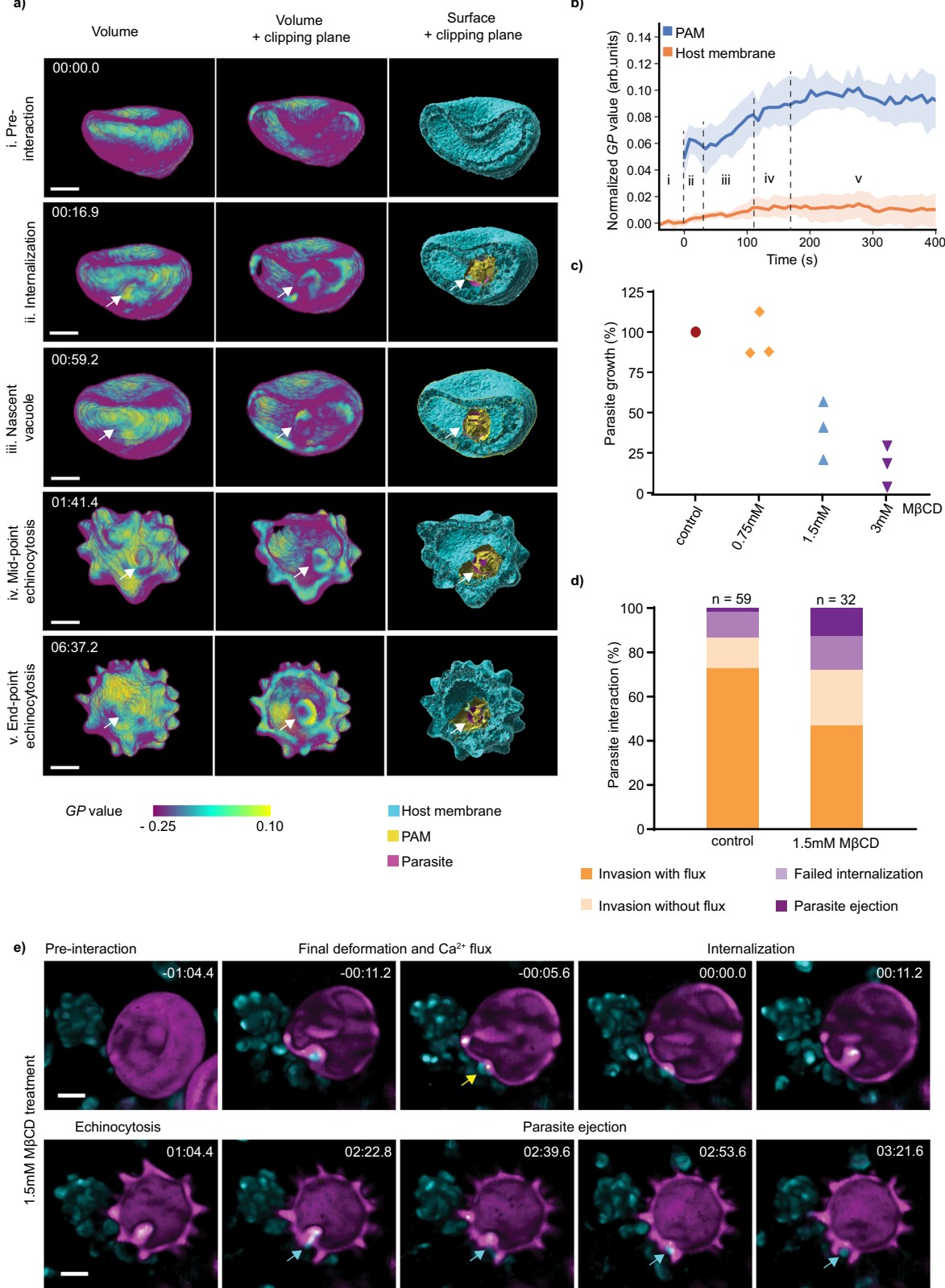

invasion and measuring the effects of inhibitors and drugs that act downstream of the PfRh5–basigin interaction[16].

Based on our observations, red blood cells begin to undergo a transition from the classical biconcave shape to an echinocyte during the internalization phase of invasion, suggesting that the process of echinocytosis begins much earlier than thought.

The origins of this shape transition are presently unknown and have been questioned for some time[15]. With the $Ca^{2+}$ flux as a reference, we noticed a striking difference in the kinetics of echinocytosis in the presence of inhibitors. In addition, we observed large membrane-based protrusions from the apex of the parasite that extended internally (i.e., with cytochalasin D) or

**Fig. 5 Erythrocyte membrane cholesterol is involved in parasite-induced host membrane remodeling for a successful invasion. a** A representative mapping of cholesterol distribution, from five independent experiments, on the erythrocyte membrane during parasite invasion using ratiometric imaging where the generalized polarization (GP) value corresponds to relative cholesterol level. GP mapping and segmented surfaces were displayed in volume view and a clipping plane was applied to highlight the parasite-associated membrane (PAM, white arrow) using IMARIS software. The time-lapse images show cholesterol enrichment at the PAM starting from the early stage of internalization through to echinocytosis. Scale bars: 2 μm. **b** Time plot for the GP value of the host membrane and the PAM throughout the four stages of invasion indicated in **a**, calculated using the surface segmentation as demonstrated in **a**, $n = 8$ invasion events. Values represent mean ±95% confidence interval. **c** Parasite growth inhibition assay on erythrocytes pre-treated with cholesterol depletion drug, MβCD, shows reduced parasite growth in consequence of the increase in drug concentration. See also Supplementary Table 1. **d** Statistics from live-imaging invasion assays on erythrocytes pre-treated with 1.5 mM MβCD show reduced invasion capacity of the parasite. **c**, **d** Graphs created with GraphPad Prism 9.0. **e** 3D volume-rendered images were displayed using the blend mode together with a clipping plane, performed with IMARIS, showing a failed invasion of an erythrocyte pre-treated with 1.5 mM MβCD in which the merozoite is ejected (cyan arrow) after internalization and echinocytosis ($n = 4$ in six independent experiments). Scale bars: 2 μm.

externally (i.e., with R1 peptide) to the host cell as it underwent echinocytosis. Their presence may account for the parasite-derived material (lipids and proteins) mixing with host lipid, resulting in membrane leaflet asymmetry. This also points to a fusion of parasite, rhoptry, and host cell membranes as this would be an energetically efficient way to deliver lipid material to the localized region of host cell during invasion[7]. Indeed, our data on RON3 suggest that proteins secreted by the rhoptry bulb after tight junction formation are associated with the PVM. Common shape transitions for erythrocytes have been well studied and moving from a classical biconcave shape to an echinocyte could be explained by a very minor imbalance in membrane leaflet surface area[48]. Indeed, it has previously been shown that an increase in the surface area of the outer leaflet of the membrane, relative to the inner leaflet of the erythrocyte membrane (as described in the bilayer couple hypothesis) or other factors such as cholesterol enrichment, can induce echinocytosis[49,50]. Coupled with the surface area of the forming PVM and early shape transition, our results lead us to hypothesize that echinocytosis, which accompanies malaria parasite invasion, is likely to be a lipid-driven process that can occur independently of PVM formation. The release of parasite lipids and proteins may therefore contribute to the local remodeling of the host membrane that becomes the principal component of the newly formed vacuole.

Despite being mostly composed of host membrane, the nascent PVM has an elevated cholesterol level compared with the rest of the erythrocyte membrane. Our data show cholesterol accumulation at the parasite-host interface from the outset of internalization. Although it could come from the invading parasite, there is no evidence that *P. falciparum* can synthesize cholesterol de novo. The reduced capacity for merozoites to invade cholesterol-depleted erythrocytes, as revealed by our invasion assays, supports the view that host membrane cholesterol has an important role in invasion. Cholesterol might be actively recruited to the PVM or there could be a mechanical redistribution of host membrane lipids during membrane deformations[30,51]. Cholesterol has an intrinsic negative curvature, which might help to stabilize the large curvatures on the host membrane during invagination[52]. The recruitment of host membrane cholesterol to the invasion site, whether a passive feedback of the lipid bilayer to alleviate mechanical strain during invagination or an active molecular response to parasite interaction, could also explain why there is selective uptake of host detergent-resistant membrane-associated proteins by the PVM[28–30,46]. Detergent-resistant membrane-associated proteins partition into detergent-resistant membranes, which are enriched in cholesterol and sphingolipids.

The final stage of PVM biogenesis involves the hemi-fusion of opposing invaginated membranes. Through the initial fusion stages of the sealing process, it is likely that the opposing membranes will occupy a mid-stage hemi-fusional state, as proposed by the stalk-pore model of membrane fusion[53], and cholesterol is often implicated in the stabilization of this state and the progression to a fully fused position[54,55]. When invasion occurs in a cholesterol-depleted erythrocyte, this is likely left unstable, resulting in the ejection of the parasite through the invasion site as osmotic conditions within the erythrocyte change. Conventional imaging technologies have been unable to resolve the details of these processes because they have lacked the necessary spatiotemporal resolution (e.g., widefield DIC or confocal) or have relied on chemical fixation, which can only provide a limited number of snapshots in time (e.g., electron microscopy). PVM sealing is followed by, what is thought to be, eventual fission as the PVM is released into the erythrocyte cytosol. The methodology described here, which is based on LLSM, demonstrates previously unseen dynamics of the internalized parasite immediately following the sealing of the PVM with a remarkable level of detail and therefore offers an opportunity to probe these mechanisms in further studies.

The plasma membrane of cells represents the first line of defense against bacterium, virus, and parasite entry. Although host cell adherence involves unique ligand–receptor interactions on the membrane, microbial pathogens share common strategies related to membrane bending, integral protein clustering, and cytoskeletal reorganization. The erythrocyte is one of the simplest biological systems for studying human pathogens. They are neither phagocytic nor endocytic. To invade erythrocytes, the molecular machinery of *Plasmodium* must actively overcome both the lipid bilayer and its underlying cytoskeleton. The volumetric information offered by this modality could expand on the emerging evidence of the role of the physical properties of the host cell in the invasion process[56]. However, the methods described, and data shown here are not exclusively relevant to *Plasmodium*, as our imaging framework promises to add new information to our understanding regarding the biophysical and molecular strategies of other microbial pathogens. Questions can now be asked on elements of invasion that are frequently overlooked in the context of therapeutic development. What are the molecular and/or mechanical pathways that govern the host membrane remodeling throughout invasion? How does the PVM seal upon completion of invasion? How does the parasitophorous vacuole detach itself from the host membrane? Data of such high quality may pave the way to answer some of these questions and initiate new avenues of drug development against a continually shapeshifting, goalpost moving, and familiar foe.

## Methods

***P. falciparum* culture**. Asexual stage 3D7 *P. falciparum* parasites were cultured in human O + erythrocytes (Australian Red Cross Blood Service) at 4% hematocrit in Roswell Park Memorial Institute (RPMI) 1640 medium (Gibco) supplemented with 26 mM 4-(2-hydroxyethyl)piperazine-1-ethanesulfonic acid (HEPES), 50 mg/mL hypoxanthine, 20 mg/mL gentamicin, 0.2% NaHCO₃, and 10% Albumax II (Gibco), as previously described[57]. In brief, cultures were grown in 30 mL petri dishes and kept at 37°C in 1% O₂, 5% CO₂, 94% N₂. Culture medium was replaced

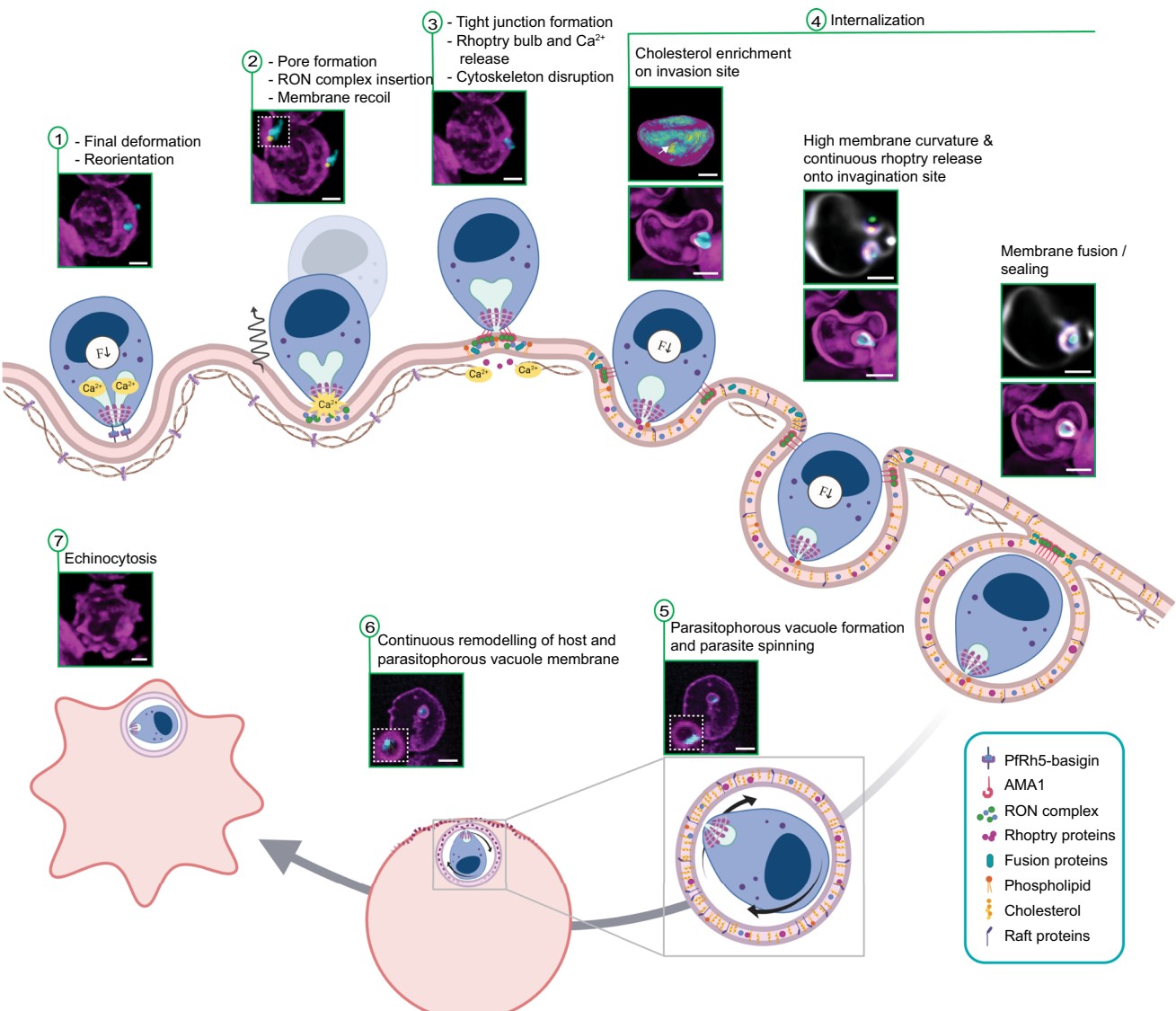

**Fig. 6 Refined model for invasion.** (1) Upon egress, the merozoite deforms a nearby erythrocyte multiple times until a reorientation of the merozoite is achieved and its apical tip is anchored to the host membrane via PfRh5–basigin interaction. (2) Following this interaction, a fusion pore is formed at the parasite-host junction, and RON complex is inserted into the erythrocyte membrane as the parasite pulls back. (3) A tight junction is then formed as the parasite binds the RON complex with AMA1. Coincidentally, the parasite releases $Ca^{2+}$, proteins, and lipids, from its rhoptry, which is followed by the disruption of the erythrocyte cytoskeleton underneath the parasite-host interface. (4) As the parasite begins to internalize, cholesterol is recruited to this site to sustain high curvature at the aperture of the invasion site. The energetic contributions of cholesterol accumulation, cytoskeleton disruption, and parasite motor engagement allow the merozoite to be completely wrapped by the remodeled host membrane. The presence of cholesterol enables membrane fusion to seal invagination orifice and facilitates host detergent-resistant membrane-associated proteins recruitment into the PVM. (5–6) Upon invasion, the parasite spins inside the vacuole and the PVM continues to be enriched in cholesterol and other parasite-derived materials. (7) Eventually, the continuous membrane remodeling induced by the parasite causes the host cell to undergo echinocytosis (created with BioRender.com). Snapshots taken from multiple live-imaging experiments. Scale bars: 2 μm.

with fresh medium every 2 days. Parasite synchronization was done by eliminating mature parasites (non-ring stages) with 5% sorbitol, as described previously[58]. In brief, the culture medium was removed and the cells were incubated in 5× volume of 5% sorbitol in a water bath at 37°C for 7 min. The sorbitol was then washed-off and fresh culture medium was added back to the synchronized culture.

**Construction of *P. falciparum* line expressing mNeonGreen-tagged RON₃.** The parasite was made using the CRISPR-cas9 system[16,59]. The strategy involves the generation of a guide plasmid and a plasmid that replaces the endogenous RON3 gene with a mNeonGreen-tagged version (the homology-directed repair or HDR plasmid). The guide plasmid was made in the pUF1-Cas9G vector, with the DNA sequence 5′-GATAGTGAAAAAGCATATGG-3′ acting as the RON3 guide. The HDR plasmid was made in three steps, with 5′ and 3′ flanks amplified from 3D7 gDNA and a recodoned RON3 sequence (GeneArt) downstream of the cas9 cleavage site fused to the 5′ flank. This was assembled in a modified p1.2 plasmid

encoding WR99210 resistance where 3xHA was replaced with 3xHA-nGreen at the 3′ end. The 5′ flank was amplified with the primers 5′-AGCTGCGGCCGCAAAAT ATAACTAAACCATCAGATC-3′ and 5′-TATTGTATCTGGACTAACCATGG-3′ while the 3′ flank was amplified with the primers 5′-AGCTCTGCAGCCAAAAG CAGATATTATATCTTTATATAAAATTGTGG-3′ and 5′-AGCTACTAGTATTC TTCTAACGTCAATACTGG-3′ (see Supplementary Table 2). Restriction enzyme-linearized HDR plasmid (50 μg) and circular guide plasmid (100 μg) were trans-fected simultaneously into E64-treated late-stage 3D7 schizonts as previously described[60]. Parasites with an integrated mNeonGreen-tagged RON3 were selected and maintained on 2.5 nM WR99210. Synchronized late-stage parasites were saponin-lysed to remove uninfected erythrocytes and parasite proteins for western blot obtained by lysis in reduced Sample Buffer.

**Immunofluorescence assay.** Synchronized schizont cultures were magnet-purified and incubated in E64 (10 μM) for 3–5 hours until fully segmented.

Parasites were passed through a 1.2 µm syringe filter to obtain merozoite suspension. Merozoites were mixed with erythrocytes in culture media supplemented with dimethyl sulfoxide or R1 peptide (100 µg/ml) and incubated at 37°C with shaking for 1 min 30 s until fixing. Parasite cultures were spun down at 2000×g and fixed with 4% paraformaldehyde and 0.01% glutaraldehyde in phosphate-buffered saline (PBS) for 30 min, permeabilized with 0.1% Triton X-100 in PBS for 25 min, and incubated in blocking solution (2% bovine serum albumin in PBS) for 1 hour at room temperature. Cell suspensions were then mounted onto 1% polyethyleneimine-treated coverslips, the excess liquid was removed and samples were mounted on slides with Vectashield containing 4′,6-diamidino-2-phenylindole (DAPI). The primary antibody used to detect the C-terminus of RON3 was rat anti-HA (3F10, Roche, 1:500) and the secondary antibody conjugated to Alexa-488 fluorophore from Thermofisher (1:1000). Wheat Germ Agglutinin 647 conjugate (Thermofisher) was used to stain the erythrocytes at 1:1000 dilution.

**3D-structured illumination microscopy.** Super-resolution three-dimensional structured illumination microscopy (3D SIM) was performed on the DeltaVision OMX-SR system (GE Healthcare) equipped with a ×60/1.42 NA PlanApo oil immersion objective (Olympus), sCMOS cameras, and 405, 488, 568, and 640 nm lasers, and 1.518 or 1.520 refractive index immersion oil. To image RON3 localization on late-stage parasites, synchronized mNeonGreen-tagged RON3 schizonts were labeled with Sir-DNA (Spirochrome) nucleus dye. The 488 nm laser was used to excite mNeonGreen and the 640 nm laser was used to excite Sir-DNA. To image RON3 localization on samples fixed upon invasion assay, the 405 nm laser was used to excite DAPI, the 488 nm to excite mNeonGreen-tagged RON3, and the 640 nm to excite the wheat germ agglutinin membrane label. Structured Illumination image stacks were constructed from 15 raw images per plane (five phases, three angles) per color channel and a z-step size of 125 nm. Images for widefield deconvolution were recorded under standard widefield excitation with a single recorded image per color channel per z-stack with a z-step size of 375 nm. Super-resolution reconstruction or widefield deconvolution and color channel alignment were performed with softWoRx 7.0 (GE Healthcare).

**Growth inhibition assay.** Packed erythrocytes were washed once in incomplete medium then resuspended back in 1 mL of incomplete medium at 1% hematocrit. The erythrocytes were either left untreated or treated with MβCD (Sigma Aldrich C4555) at 0.75 mM, 1.5 mM, and 3 mM for 30 min at 37°C. The treated erythrocytes were then washed twice and each untreated and treated cell with different concentrations was left as a 10 µL pellet in Eppendorf tubes. Free merozoites were harvested from synchronized 3D7 culture, as previously described[61]. In brief, schizonts were stalled from egress by adding 10 µM E64 (Thermo Scientific) in the culture medium. After 3 h, schizonts were washed from E64 and resuspended in complete medium. Schizonts were then loaded to a syringe and pushed through a 1.2 µm filter (Sartorius) to obtain free merozoites. Equal amount of free merozoites were added to the untreated and pre-treated erythrocytes. The samples were then put on a shaker heated to 37°C for erythrocyte invasion to occur. After 25 min, the samples were spun down and the supernatant discarded to exclude the merozoites that had not invaded. Each sample was then resuspended in complete medium at 5% hematocrit. Each sample was loaded in triplicate on a round-bottom 96-well plate (Falcon 353077) with 50 µL volume per well and the plate was kept in culture conditions. The next day, parasites were stained with ethidium bromide (EtBr, Bio-Rad) at 1:1000 dilution in DPBS for 20 min and ~100,000 erythrocytes were counted for each sample with flow cytometer (Becton Dickinson FACSCalibur) using 488 nm laser to excite EtBr on stained parasites. The data were analyzed using FlowJo (Tree Star Inc) where erythrocytes were gated based on side scatter and forward scatter, then the parasitemia was determined based on EtBr-positive cells.

**Erythrocyte preparation for live microscopy invasion assays.** Erythrocytes were stained differently to suit each experiment using a combination of a calcium reporter (Fluo-4AM) and a membrane marker (PKH26 or Di-4-ANEPPDHQ). For experiments with high acquisition rate (i.e., Figs. 1 and 2), the erythrocytes were stained with 0.5 µM PKH26 (Sigma Aldrich MINI26-1KT) in Diluent C for 5 minutes at 1.5% hematocrit. For Ca²⁺ flux study (i.e., Figs. 3 and 4), the erythrocytes were stained with 10 µM Fluo-4AM (Invitrogen F14201) at 0.5% hematocrit in phenol red-free RPMI-HEPES supplemented with 0.2% NaHCO₃ and 5 mM sodium pyruvate (Gibco 11360070). After 1 hour, 1.5 µM Di-4-ANEPPDHQ (Invitrogen D36802) was added to the cell suspension for another hour of staining. For cholesterol study (i.e., Fig. 5), the erythrocytes were stained with 1.5 µM Di-4-ANEPPDHQ at 0.5% hematocrit for 1 h in phenol red-free RPMI-HEPES supplemented with 0.2% NaHCO₃. For the cholesterol depletion study, the erythrocytes were pre-treated with 1.5 mM MβCD at 1% hematocrit for 30 minutes in room temperature, and then washed twice before being stained with Di-4-ANEPPDHQ. All staining was done at 37°C and the stained erythrocytes were washed for three times and resuspended in phenol red-free RPMI-HEPES supplemented with 5 mM sodium pyruvate.

**Parasite preparation for the live microscopy invasion assays.** Late-stage parasites (schizonts) were isolated from synchronized culture with magnetic-

activated cell sorting magnet separation column (Miltenyi Biotec). Purified schizonts obtained from the magnet purification were put back to culture conditions and some of the schizonts were treated with 2 µM Compound 1 (pyrrole 4-[2-(4-fluorophenyl)-5-(1-methylpiperidine-4-yl)-1H-pyrrol-3-yl] pyridine) to stall them from egress for later imaging sessions. For each imaging session, a batch of schizonts was stained with a mitochondrial dye (25 nM–100 nM of Mitotracker Green, Mitotracker Red, or Mitotracker Deep Red) (Invitrogen M7514, M22425, M22426) for 30 minutes and then washed-off from the dye and Compound 1 (if there is any).

**Sample mounting.** The sample was imaged in phenol red-free complete medium. For calcium experiments, the imaging medium was supplemented with 5 mM sodium pyruvate, 10 µM Trolox (Santa Cruz 53188-07-1) and 0.25 mM CaCl₂. For experiments involving inhibitors, either 100 µg/mL R1 peptide (China Peptides), 1 µg/mL cytochalasin D (Sigma Aldrich C8273), or 10 µM TRAM34 (Sigma Aldrich T6700) were added to the imaging medium. For confocal microscopy, 200 µL of imaging medium and 30 µL of stained erythrocytes per well were loaded to eight-well plate (Ibidi 80826). Then, 5–10 µL of stained schizonts were added to the stained erythrocytes in the first well right before imaging. For LLSM, an acid-washed 5 mm round glass coverslip (Warner Instruments CS-5R) was attached to the bottom of each well before loading 200 µL of phenol red-free RPMI-HEPES and 30 µL of stained erythrocytes to the well. Then, 5–10 µL of stained schizonts were gently added to the top of the coverslip and left to settle on the coverslip for 15 minutes. A tweezer was used to attach the coverslip to the sample stage, then the coverslip was embedded in the microscope bath filled with 8 mL of imaging medium.

**Lattice light-sheet microscopy.** For all LLSM experiments, a custom home-built system was used, constructed as outlined in ref. [33] as per licensed plans kindly provided by Janelia Farm Research campus. Excitation light from either 488 nm, 561 nm, 589 nm, or 642 nm diode lasers (MPB Communications) was focused to the back aperture of a 28.6 × 0.7 NA excitation objective (Special optics) via an annular ring of 0.44 inner NA and 0.55 outer NA providing a light sheet 10 µm in length. Fluorescence emission was collected via a ×25 1.1 NA water dipping objective (Nikon) and detected by either one or two sCMOS cameras (Hamamatsu Orca Flash 4.0 v2). In the instances of dual-channel imaging detected fluorescence was split using either a 561 nm or 594 nm dichroic (Semrock). Imaging of membrane cholesterol was performed using the dual-channel custom home-built LLSM. Emitted fluorescence was collected via a 594 nm dichroic before passing through a 405/488/594 nm multi-band filter (Chroma) and 540/80 nm filter providing detection windows of 500–580 nm and 610–710 nm. A liberally coated fluorescence bead slide was imaged to provide reference for registration of the two channels. Registration was performed using LLSpy and an Affine transformation. Data were deskewed for *GP* processing but not deconvolved. For imaging Mitotracker Green in combination with PKH26 525/50 nm and 405/488/561/633 multi-band filter sets were used, respectively. For imaging Fluo-4AM, Mitotracker deep red and Di-4-ANEPPDHQ 525/50 nm and 405/488/561/633 multi-band filter sets were used respectively. All data were acquired in an imaging chamber (Okolabs) set to 36°C and 5% humidified CO₂. For all experiments, Point Spread Functions (PSFs) were measured using 200 nm Tetraspeck beads on the surface of a 5 mm coverslip. Data were deskewed and deconvolved using LLSpy, a Python interface for processing of LLSM data. Deconvolution was performed using a Richardson-Lucy algorithm using the PSFs generated for each excitation wavelength.

**Confocal microscopy.** Live-cell confocal microscopy was undertaken on a Leica SP8 resonant scanning confocal and ×63/1.4 NA oil immersion objective. Cells were maintained at 36 °C in a humidified chamber with 5% CO₂. We used the Z-Galvo function and AOBS to acquire very fast z-stacks together with an 8 kHz resonant scanner. Channels were usually acquired simultaneously. Fluo-4AM (500–550 emission) and Di-4-ANEPPDHQ (500–550 nm and 620–750 nm dual-band emission), were excited using the 488 nm laser line. Mitotracker Red (580–620 nm emission) or Deep Red (620–750 nm) were excited using the 561 nm and 633 nm laser lines, respectively. All data were captured using either a PMT or HyD detector. Images were captured in line mode with 2× averaging. Acquisitions typically continued for ~5 minutes after an egress event. Laser levels were kept as low as possible at <2% and transmitted light images were captured concurrently. Time-lapse videos were recorded with an 8 µm z-stack (1 µm step size) at 0.96 seconds per stack (eight slices per z-stack). The recording was started as soon as schizonts egressed or just prior to egress. In some experiments, we increased the number of slices to cover the full cell volume and meet Nyquist criteria, but as a tradeoff, a reduced frame rate was applied.

**Validation of Di-4-ANEPPDHQ as a cholesterol reporter.** Packed erythrocytes were washed once in incomplete medium then resuspended back in 1 mL of incomplete medium at 1% hematocrit. The erythrocytes were then either left untreated or treated with MβCD, MβCD conjugated with cholesterol, or sphingomyelinase, in doubling concentrations from 0.75 mM to 3 mM, 3.75 µM to 15 µM, and 2.5 mU/mL to 10 mU/mL, respectively. The erythrocytes were incubated with MβCD and MβCD-cholesterol for 30 min and with sphingomyelinase

for 15 min at 37°C. Untreated and treated erythrocytes (washed twice after treatment) were then stained with Di-4-ANEPPDHQ and prepared for imaging on the LLSM, as previously described. The fluorescence emissions were split and captured with the dual-channel detection set up for ratiometric imaging of membrane cholesterol, as described earlier.

**Parasite-associated host membrane (PAM) segmentation.** We analyzed the kinetics of invasion by tracking the formation of the PVM over time. This was performed by plotting the volume of the membrane that was in contact with the parasite channel at each time point. The analysis was performed using IMARIS (Versions 8.3 - 9.5, Bitplane) with both the IMARIS XT and Tracking modules. We first created a surface on the parasite using smoothing and an absolute intensity setting. The thresholding was either automated or in some cases adjusted manually to ensure boundaries of the parasite channel were included. Surfaces were split (1 μm) and a quality filter used (usually with an automatic threshold). Parasites were then selected and all pixels outside of the parasite channel were set to a value of 0 and a new channel created (called parasite channel). Next, a surface was created on the erythrocyte membrane using an absolute intensity setting with a threshold of 30. In some cases, a manual threshold was performed, and surfaces were split (0.2 μm) and automated quality filter used (this was called the erythrocyte channel). Next, a filter was applied to the erythrocyte channel (i.e., surfaces created on the erythrocyte membrane) using the function "Intensity max" of the newly created parasite channel. This method enabled those surfaces from the erythrocyte channel, which were in contact with the parasite channel to be highlighted. Those surfaces were then duplicated and unified and volumes exported to Microsoft Excel for the PAM analysis.

**Surface area and volume measurements.** The surface area measurement of the erythrocytes before, during, and after invasion was performed using a custom-written ImageJ/FIJI macro for semi-automated analysis, which makes extensive use of the LimeSeg plugin[35]. Segmentation of the membrane channel required defining an initial region within the cell to be used as a seed point and multiple parameters for the LimeSeg algorithm to perform accurately. These regions and parameters were set at the beginning of each time-lapse and checked for accuracy. The macro would then step through the time-lapse performing the segmentation at each time point. The segmentation was monitored as the macro progressed and necessary alterations of the segmentation parameters were done accordingly. For PVM and host membrane surface area calculations, five time points were averaged before and after sealing of the PVM. Reduced volume was calculated as the ratio of the volume of the erythrocyte to that of a sphere with an identical surface area by the following relationship:

$$V_R = \frac{6\sqrt{\pi}V}{SA^{3/2}} \quad (1)$$

where $V_R$ is the reduced volume, $V$ is the measured volume and $SA$ is the measured surface area.

**3D Gaussian curvature analysis.** Erythrocyte segmentation and surface triangulation were performed using LimeSeg (see above/below). The obtained triangulated 3D surface meshes, consisting of vertices and triangles, were imported into MATLAB (Mathworks, MA, USA) using the "plyread" script. The "patch curvature" script was then used to calculate the Gaussian curvatures of the triangulated meshes. In brief, for each of its vertices, the triangulated mesh is rotated such that the normal of the vertex becomes [−1 0 0], and the local mesh topography around the vertex becomes a function of two coordinates. Then, a quadratic function is fitted to the vertex neighborhood and the eigenvectors and eigenvalues of the hessian matrix are used to calculate the Mean, Gaussian, and principal curvatures. As opposed to the Mean curvature, which depends on the space the surface of interest is embedded in, the Gaussian curvature is intrinsic, i.e., independent of the coordinate system. Erythrocyte surface regions with negative Gaussian curvatures are shown in red and regions with positive Gaussian curvatures in blue, with color scale limits indicated.

**GP processing.** For all GP data, two images were acquired separately to be combined into one ratiometric image. All GP processing was performed using custom-written ImageJ/FIJI macro, adapted from ref. [43], and was capable of batch processing data sets. The two images, corresponding to the wavelength range 500–580 nm and 610–710 nm, were combined via the following relationship:

$$GP = \frac{I_{500-580} - I_{610-710}}{I_{500-580} + I_{610-710}} \quad (2)$$

Images were auto-thresholded using an Otsu threshold prior to GP calculation to eliminate skewed GP values owing to the inherent difference in intensity between the channels. The generated GP image stack ranged from $-1 < GP < 1$ corresponding to low to high cholesterol values. GP images were pseudo-colored with a Rainbow RGB look-up table for clarity of demonstration (Supplementary Figure 7). Following GP processing the PVM and host membrane were segmented using the PAM segmentation method.

**Reporting summary.** Further information on research design is available in the Nature Research Reporting Summary linked to this article.

## Data availability
All data used in this study are available upon reasonable request. Source data are provided with this paper.

## Code availability
All customized ImageJ macros or other forms of data code used during this study are available upon reasonable request.

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

## Acknowledgements

We thank Brad Sleebs for providing Compound 1 and Kitsanapong Reaksudsan for assistance in maintaining parasite cultures. We also thank Nina Tubau for assistance with data processing. The lattice light-sheet referenced in this research was used under license from Howard Hughes Medical Institute, Janelia Research Campus. We acknowledge the Australian Red Cross Blood Service for providing blood. This work was supported by an EMBO Long Term Fellowship ALTF 793-2016 and Sir Henry Wellcome Fellowship 206515_Z_17_Z to M.P. This work was also supported by grants from the National Health and Medical Research Council (NHMRC), APP1177431 to D.M, and NHMRC APPs 1121178, 1092789 and 1117288 to A.F.C.

## Author contributions

C.E., N.D.G, and P.M. performed the LLSM experiments. C.E., M.P., and P.M. optimized sample preparation protocols for live cell imaging. C.E., J.K.T., M.P., and J.H. prepared parasites. C.E., D.M., and M.P. performed the growth inhibition assay. M.P. performed the immunofluorescence assay. T.T. generated the mNeonGreen-tagged RON3 parasite line. M.M. performed 3D SIM. N.D.G. and L.W.W. custom-built the LLSM. N.D.G. and L.W.W. established the data processing and analysis pipelines. N.D.G., C.E., K.L.R., D.K., M.M., and L.W.W. performed data analysis. K.L.R., N.D.G., C.E., and A.F.C. conceived and designed experiments. K.L.R., A.F.C., and M.B. supervised the work. C.E., N.D.G., and K.L.R. wrote the paper.

## Competing interests

The authors declare no competing financial interests.
