## [Peer Review File · Nature Communications]

Reviewer Comments, first round –

Reviewer #1 (Remarks to the Author):

Geoghegan et al. "4D analysis of malaria parasite invasion offers new insights into erythrocyte membrane remodeling and origins of the parasitophorous vacuole" offers thought provoking new insights into the process of malaria parasite invasion into red blood cells. Using lattice light sheet microscopy, the authors are able to resolve the invasion process with unprecedented time and spatial resolution. Highlights of the paper are that the authors show that the nascent parasitophorous vacuole is apparently derived completely from the red cell plasma membrane. Also, for the first time, the parasite can be seen rotating in the parasitophorous vacuole. The authors quantify the physical parameters of the membrane in determining gaussian curvature and generalized polarization (GP). With those measurements the authors speculate about the role of lipid sorting and curvature stress for the invasion process.

A problem with the manuscript is that the measurement of the biophysical quantities GP and curvature are not motivated, but rather justified post hoc: it appears that curvature was determined without spelling out a clear hypothesis. There are lines of argument that could then lead to the critical experiments, alternative hypotheses, etc. Perhaps the authors can find a way to calculate the forces required to deform membranes the way the parasite does. I think this would then also allow an easier transition to the GP measurement. At the moment there is only a loose line of argumentation on the basis that GP seems to correlate with cholesterol content, but GP is a rough measure of the behavior of an exogenous dye and GP is affected by many cellular and membrane factors, including the cytoskeleton. Can you please discuss alternative parameters measured by GP, other than cholesterol? The conclusion that cholesterol makes the membrane flexible enough to deform for invasion and invasion is impossible without it is only one possible explanation and insufficient controls for alternate hypotheses are examined and tested. A calculation of the required energy for deformation should be used to argue that this is indeed reasonable to expect. After all, treating the parasite with beta-CD takes away lipid area that simply may be necessary for the parasite to form the vacuole, especially given that the RBC cortex doesn't deform seem to deform much during invasion. I also like to note that the usage of the raft concept seems antiquated. I think the word raft should be replaced by a term such as "local enrichment of cholesterol" as the raft concept is unlikely to hold in its originally postulated form.

Some minor points:

In all movies a time stamp and scale bar should be shown to give a sense of scale and dynamics.

The authors could exploit the isometric pixels to find a "non-orthogonal" image plane (e.g. the oblique slicer in imaris does that) that shows features clearly. For example, in figure 3 I don't see the membrane deformation the way the figure is made.

The authors show that no membrane area is added from the parasite during invasion, yet later they seem to claim that lipids are transferred from the parasite. Those two observations seem to be contradictory. Can you please resolve the contradiction?

In line 152 the authors speculate the surface to volume change resulting from the loss of membrane from the PVM formation is the onset of echinocytosis. However, the authors also show that when invasion is prevented with cytochalasin D echinocytosis is still happening. These two observations seem to contradict each other.

Line 389: I believe instead of "involves the fusion" it should read "involves the hemi-fusion"

Reviewer #2 (Remarks to the Author):

This manuscript by the Rogers lab investigates erythrocyte infections by *Plasmodium falciparum* through novel multidimensional lattice light sheet (LLS) imaging and quantitative image analysis. The consequent usage of LLS microscopy allowed to clarify an impressive number of issues that have been discussed for a long time by the malaria research community: This includes the origin of the membranes forming the parasite containing vacuole, the role of Ca²⁺ signaling, the involvement of factors released from the parasite (such as Ron3), the involvement of the actin-myosin motor, cholesterol accumulation at the vacuole, and the events that underlie the events of echinocytosis. I was very impressed by the power of the rapid LLS imaging, and the elegance of the analytical tools developed by and applied by the authors for quantification of the complex multidimensional data. LLS microscopy offers a huge promise for the study of the dynamic host-parasite interactions as it provides unprecedented 3D temporally resolved data with very low phototoxicity. This could not be achieved by other methods, and the presented work has the potential to become a landmark paper for other "sensitive" host-pathogen studies, even outside the field of parasites. For example, bacterial infection is often halted by too much light exposure, and LLS microscopy could become a gamechanger for such studies. I am also intrigued that questions of mechanobiology can be tackled with the approach. In summary, the authors exploit LLS microscopy in the best possible way to solve longstanding controversies during malaria-erythrocyte interactions. This important work is highly relevant to all scientists working on host-pathogen interactions, and who are interested in their quantitative understanding.

I have a few minor comments that could help to underline the exceptional nature of the work:

The authors should use a time stamp in all of their movies, so scientists realize how rapid the volumes have been imaged.

The genetic fluorescent tagging of RON3 may alter its function and localization. This could be potentially addressed through a control experiment labeling RON3 by indirect immunofluorescence. However, due to the short-lived transient nature of the events, such experiments are risky and could potentially fail. Therefore, I would suggest that the authors add data in case they have some or are convinced to obtain it. Or they should simply discuss the potential limitations of the chosen tagging approach.

The scheme in figure 6 is very well designed. It is obvious that the parasite is not shrinking between step 5 and the following steps, but the authors could indicate that they zoomed out at this step.

Reviewer #3 (Remarks to the Author):

The authors provide a superb imaging study on merozoite invasion of red blood cells that reveals among much more the enrichment of cholesterol at the invasion site and use lattice light sheet imaging for the first time in the field.

The only minor critique I have is textual:

page 1, line 20: please delete long-standing

1,21: what is a computational framework? rephrase or just delete?

2,35: delete deadliest infectious

3,52: there is no such thing as a leading vaccine candidate, please delete

4,71: set the bedrock, please rephrase

4,80: please delete for the first time

the authors could include a reference to Kariuki et al Nature 2020 in their discussion as their technique could also be used for such studies, I presume.

“4D analysis of malaria parasite invasion offers new insights into erythrocyte membrane remodeling and origins of the parasitophorous vacuole”

Reviewer Comments

Reviewer #1:

Geoghegan et al. “4D analysis of malaria parasite invasion offers new insights into erythrocyte membrane remodeling and origins of the parasitophorous vacuole” offers thought provoking new insights into the process of malaria parasite invasion into red blood cells. Using lattice light sheet microscopy, the authors are able to resolve the invasion process with unprecedented time and spatial resolution. Highlights of the paper are that the authors show that the nascent parasitophorous vacuole is apparently derived completely from the red cell plasma membrane. Also, for the first time, the parasite can be seen rotating in the parasitophorous vacuole. The authors quantify the physical parameters of the membrane in determining gaussian curvature and generalized polarization (GP). With those measurements the authors speculate about the role of lipid sorting and curvature stress for the invasion process.

A problem with the manuscript is that the measurement of the biophysical quantities GP and curvature are not motivated, but rather justified post hoc: it appears that curvature was determined without spelling out a clear hypothesis. There are lines of argument that could then lead to the critical experiments, alternative hypotheses, etc. Perhaps the authors can find a way to calculate the forces required to deform membranes the way the parasite does. I think this would then also allow an easier transition to the GP measurement.

We thank the reviewer for their comments and insights into the presented work. We agree that the motivation for measuring the GP values and curvature could be clearer within the narrative of the paper.

Measurement of curvature:

The inclusion of the curvature measurement was primarily as a means of highlighting the clear visual differences during the invasion process both in the presence and absence of the functioning tight junction using R1 peptide. The lower degree of curvature seen when invasion is inhibited suggests that the large deformations alone are insufficient for successful invasion and that the underlying cytoskeletal disruption and tight junction formation are key in minimizing the energy required for successful invasion. The force exerted by the parasite in both cases would likely be similar due to most of its energy for invasion originating from its acto-myosin motor. The distinct differences in curvature between the inhibited and control conditions indicate that the host cell is less deformable due to the lack of a functioning tight junction and points to a role for the AMA1-RON2 interaction in disrupting the underlying cytoskeleton. This is highlighted in lines 207-216.

We agree with the reviewer that the ability to measure the imposed forces by the parasite using this curvature data would be of great interest and would extend the capacity of the present technique within the field. While the calculations of these forces initially seem

trivial, we find them to be computationally complex once extended to the 4D data that we currently present. Measuring stretching forces on model membranes using shape as a reporter has been previously demonstrated in 2D but has yet to be extended to 3D data [1]. The requirement would be to develop a new analytical method based on 3D surface curvature data coupled to derivations of the Helfrich model for membrane forces. An alternative approach may be to combine Atomic Force Microscopy (AFM) measurements with light sheet imaging, such as that demonstrated very recently [2], to create a generalized correlation between induced curvatures whilst directly measuring the imposed forces on red blood cell membranes. However, with either of these proposed approaches the assumption is that the host cell remains in a steady state and they would not consider any remodeling of the host cell membrane/cytoskeleton. It is known that during invasion the parasite disrupts the underlying cytoskeleton resulting in a change to the localized membrane tension. A biophysical model for invasion was presented in 2014 which proposed four major energetic contributors to the invasion process: adhesion strength of the parasite to the host cell, membrane bending energy, membrane tension and line tension [3]. By developing this technique, we would be able to empirically evaluate the proposed biomechanical model by assessing each component of energetic contributors [3]. This would require the generation of mutant parasite lines or knock out lines which can target tight junction formation, cytoskeletal disruption, motor force generation in addition to experiments where the host membrane composition is also altered. This would be a valuable study in its own right. To this end we believe that to develop the method and interrogate more closely the biophysical mechanisms of invasion would be beyond the scope of the current paper.

Measurement of GP:

Our motivation for extending the imaging technique to involve environmentally sensitive dyes was driven by the observed localized membrane disruption when invasion is inhibited and by the conclusion that the PVM is predominantly formed from host membrane material. In addition, it has previously been reported that the nascent PVM is rich in detergent resistant membranes and more ordered domains. This indicates that there is a divergence between the composition of the nascent PVM and the invaded host cell. When and how the PVM is remodeled remains an unanswered question and we hypothesized that this process is initiated during the invasion process. We were motivated to implement the functional imaging approach to see if it was possible to determine the dynamics of this process and at what point the forming PVM demonstrated deviations from the host membrane composition. At the suggestion of the reviewer, we have attempted to clarify this motivation by adding the following:

Page 13, lines 292-302: *“While our data strongly suggests that the PVM is largely formed from host membrane material, there is evidence in the literature that the nascent PVM is rich in detergent resistant membrane associated proteins and cholesterol. This could be linked to the high membrane curvatures produced during invasion, which might lead to the mechanical redistribution of host membrane lipids and their associated proteins or it may be due to the active transport of these lipid domains into the newly forming PVM. This suggests*

there is early remodeling of the PVM during the invasion process. Our observation that there is localized membrane disruption in the presence of invasion inhibitors and that echinocytosis may be a lipid driven process suggests that the remodeling process begins early after the commencement of internalization. To address this question, we measured local changes to the membrane composition of the forming PVM during invasion. To temporally measure changes in lipid composition....”

At the moment there is only a loose line of argumentation on the basis that GP seems to correlate with cholesterol content, but GP is a rough measure of the behavior of an exogenous dye and GP is affected by many cellular and membrane factors, including the cytoskeleton. Can you please discuss alternative parameters measured by GP, other than cholesterol?

GP value has traditionally been used as a proxy for lipid order as it infers a measure of the relative hydration level of the dye molecules in the bilayer. We found that, in the context of red blood cells, that the GP value responded reliably to changes in cholesterol, as opposed to lipid order in general, by using different membrane altering agents (Supplementary Figure 6). It was also recently shown that Di-4-ANEPPDHQ does not respond directly to lipid order in a way that other solvatochromic dyes do, such as Laurdan. The authors of this study conclude that, in the context of model membranes, that the GP value of di-4-ANEPPDHQ is altered through a direct interaction with cholesterol itself. We agree with the reviewer that there is likely to be more ambiguity in a more complex environment, such as a red blood cell membrane, as opposed to a model membrane of limited lipid species. To reflect this, we have added the following information:

Page 14, lines 305-308: *“Solvatochromic dyes, such as Di-4-ANEPPDHQ, are often used as a proxy for lipid order by reporting the relative hydration level of the dye molecules in the bilayer. Recent investigations have, however, found that the response of Di-4-ANEPPDHQ is more specific to cholesterol itself rather than a measure of lipid order when compared with the well characterized probe Laurdan”.*

The conclusion that cholesterol makes the membrane flexible enough to deform for invasion and invasion is impossible without it is only one possible explanation and insufficient controls for alternate hypotheses are examined and tested. A calculation of the required energy for deformation should be used to argue that this is indeed reasonable to expect. After all, treating the parasite with beta-CD takes away lipid area that simply may be necessary for the parasite to form the vacuole, especially given that the RBC cortex doesn't deform seem to deform much during invasion.

We agree with the reviewer that the conclusion that cholesterol makes the membrane flexible enough to deform for invasion and is essential is insufficiently explored in this study. At the point of invasion, the underlying RBC cytoskeleton is severely disrupted as shown previously with a clearance of actin and other cytoskeletal elements at the invasion site [4]. It is likely that this is the more critical element in reducing the energy required to induce the large deformations at the invagination site rather than sequestration of cholesterol. We now propose that cholesterol is more likely increased at the invasion site due to its intrinsic

negative curvature and ability to stabilize the larger curvatures seen during successful invasion. We also emphasize the association with detergent-resistant membrane proteins which are known to be enriched in the PVM soon after invasion.

To clarify this we have removed the following on Page 15, line 343 '*... which would reduce the energy needed for internalization. It would also help to alleviate the strain on the growing PVM and help stabilize the structure*'.

I also like to note that the usage of the raft concept seems antiquated. I think the word raft should be replaced by a term such as "local enrichment of cholesterol" as the raft concept is unlikely to hold in its originally postulated form.

The word 'raft' has been replaced by 'detergent-resistant membrane associated proteins' or 'cholesterol enriched microdomains' where applicable.

Some minor points:

In all movies a time stamp and scale bar should be shown to give a sense of scale and dynamics.

Time stamps and scale bars have been added to all supplementary movies.

The authors could exploit the isometric pixels to find a "non-orthogonal" image plane (e.g. the oblique slicer in imaris does that) that shows features clearly. For example, in figure 3 I don't see the membrane deformation the way the figure is made.

At the suggestion of the reviewer, we have used the oblique slicer in Imaris to present a non-orthogonal view of the data presented in Figure 3. New images have been provided in the supplementary data which have now allowed us to show more clearly the membrane deformation, which is closely followed by Ca²⁺ flux and membrane recoil, right before the commencement of internalization in untreated red blood cells (Supplementary Figure 3a). We also provide two examples for the R1 peptide treated cells, because they show in addition to membrane deformation and membrane recoil, the formation of membrane tethers (See Supplementary Figure 3b). These images show very nicely the extent of deformation. Cytochalasin D treated cells are also included to show the effect of cytochalasin D, where membrane deformation occurs with reduced intensity (Supplementary Figure 3c).

The authors show that no membrane area is added from the parasite during invasion, yet later they seem to claim that lipids are transferred from the parasite. Those two observations seem to be contradictory. Can you please resolve the contradiction?

It has previously been shown that the parasite releases lipid rich content from its rhoptries during the invasion process. This has been seen by various techniques such as electron microscopy and super-resolution microscopy under various conditions. There remains the question as to their role in the invasion process. Our results measuring the surface area show that the overall majority of the newly formed PVM can be accounted for by the loss in erythrocyte surface area. However, this is only within the measurement resolution of the imaging technique and cannot rule out some contribution from the rhoptry lipids. As a

result, we hypothesize that the parasite lipid contributes a more functional role in the forming PVM and continues to contribute to its remodeling downstream of PVM sealing. We agree with the reviewer that this may appear as a contradiction due to a lack of clarity in this conclusion. To correct this, we have added the following.

Page 7, lines 155 – 158 *“This data implies that the parasite derived lipids likely play a more functional role during the invasion process. We therefore hypothesize that the parasite derived lipid material is important for further remodeling of the nascent PVM during invasion and downstream of the vacuole sealing.”*

In line 152 the authors speculate the surface to volume change resulting from the loss of membrane from the PVM formation is the onset of echinocytosis. However, the authors also show that when invasion is prevented with cytochalasin D echinocytosis is still happening. These two observations seem to contradict each other.

We have clarified this on Page 13, line 279-289 *“Together with the surface area and reduced volume data, our results therefore suggest that echinocytosis is triggered by parasite induced membrane remodeling, either involving the ejection of rhoptry bulb components into the erythrocyte membrane, which follows the Rh5-basigin interaction 15, or the formation of PVM at the completion of internalization. This is demonstrated under normal conditions and also when invasion is blocked downstream from the PfRh5-basigin interaction, such as in the presence of cytochalasin D and R1 peptide. Our observations that echinocytosis occurs more rapidly in MBCD treated cells may be relevant to the perturbed membrane composition, which could render cells more sensitive if additional lipid is then integrated into the membrane bilayer. On the other hand, echinocytosis is both delayed and prolonged when cells are treated with cytochalasin D, which could be linked to the release of rhoptry bulb material into the erythrocyte membrane beyond the tight junction.”*

Line 389: I believe instead of “involves the fusion” it should read “involves the hemi-fusion”. This has been changed.

Reviewer #2:

This manuscript by the Rogers lab investigates erythrocyte infections by Plasmodium falciparum through novel multidimensional lattice light sheet (LLS) imaging and quantitative image analysis. The consequent usage of LLS microscopy allowed to clarify an impressive number of issues that have been discussed for a long time by the malaria research community: This includes the origin of the membranes forming the parasite containing vacuole, the role of Ca²⁺ signaling, the involvement of factors released from the parasite (such as Ron3), the involvement of the actin-myosin motor, cholesterol accumulation at the vacuole, and the events that underlie the events of echinocytosis. I was very impressed by the power of the rapid LLS imaging, and the elegance of the analytical tools developed by and applied by the authors for quantification of the complex multidimensional data. LLS microscopy offers a huge promise for the study of the dynamic host-parasite interactions as it provides unprecedented 3D temporally resolved data with very low phototoxicity. This could not be achieved by other methods, and the presented work has the potential to become a landmark paper for other “sensitive” host-pathogen studies, even outside the field

of parasites. For example, bacterial infection is often halted by too much light exposure, and LLS microscopy could become a gamechanger for such studies. I am also intrigued that questions of mechanobiology can be tackled with the approach. In summary, the authors exploit LLS microscopy in the best possible way to solve longstanding controversies during malaria-erythrocyte interactions. This important work is highly relevant to all scientists working on host-pathogen interactions, and who are interested in their quantitative understanding.

I have a few minor comments that could help to underline the exceptional nature of the work:

The authors should use a time stamp in all of their movies, so scientists realize how rapid the volumes have been imaged.

A time stamp has been added to all supplementary movies.

The genetic fluorescent tagging of RON3 may alter its function and localization. This could be potentially addressed through a control experiment labeling RON3 by indirect immunofluorescence. However, due to the short-lived transient nature of the events, such experiments are risky and could potentially fail. Therefore, I would suggest that the authors add data in case they have some or are convinced to obtain it. Or they should simply discuss the potential limitations of the chosen tagging approach.

Previous studies have shown that knock-out of RON3 produces a lethal phenotype [5]. This suggests it is highly unlikely that tagging RON3 with mNeonGreen has altered its function or localization pattern as we find that these parasites develop normally without any evidence of an abnormal phenotype. It is not clear if RON3 has an important role in the invasion process. However, using this approach, we are able to determine the timing of RON3 release into the vacuolar space around the parasite during invasion. While we don't have sufficient resolution to determine if the protein is localized in the membrane, this data together with two other lines of evidence suggest that it is likely associated with the membrane. Firstly, we show the expected localization of the mNeonGreen tagged protein using 3D-SIM in the rhoptry body of merozoites just prior to egress. In addition, we have captured images of cell preparations fixed 90 seconds and 10 minutes after invasion using both widefield deconvolution microscopy (WFD) (Supplementary figure 5a and 5b) and 3D-SIM (data not shown). Our data shows that the protein is associated with the parasitophorous vacuole membrane at the completion of invasion (Supplementary Figure 5). Furthermore, we also include images showing that mNeonGreen is aberrantly released on to the host cell membrane when it is in the presence of the invasion inhibitor, R1 peptide (Supplementary Figure 5c). A reference to this figure is provided in the main document – Lines 269-271.

The scheme in figure 6 is very well designed. It is obvious that the parasite is not shrinking between step 5 and the following steps, but the authors could indicate that they zoomed out at this step.

Step 5 and step 6 of Figure 6 have been edited to indicate that the cell is zoomed-out from step 5 to step 6.

Reviewer #3:

The authors provide a superb imaging study on merozoite invasion of red blood cells that reveals among much more the enrichment of cholesterol at the invasion site and use lattice light sheet imaging for the first time in the field.

The only minor critique I have is textual:

Page 1, line 20: please delete long-standing
Done.

1,21: what is a computational framework? rephrase or just delete?
Rephrased to 'an approach'. Page 1 Line 21

2,35: delete deadliest infectious
This has been deleted.

3,52: there is no such thing as a leading vaccine candidate, please delete
This has been deleted.

4,71: set the bedrock, please rephrase
Rephrased to 'are indications of'.

4,80: please delete for the first time
Done.

The authors could include a reference to Kariuki et al Nature 2020 in their discussion as their technique could also be used for such studies, I presume.

We have added this reference to the Discussion on page, line 440.

Additional Edits:

In addition to the changes in response to the reviewers comments we have also adjusted the following:

- Colour changes to Fig 5a and supplementary movie 9. The colour scale (look up table) has been changed from Rainbow to a custom look up table as the rainbow look up table is inaccessible for colourblind readers.
- An additional example has been added to Supplementary movie 7

[1] H. J. Lee, E. L. Peterson, R. Phillips, W. S. Klug, and P. A. Wiggins, "Membrane shape as a reporter for applied forces," 2008. Accessed: Jun. 04, 2019. [Online]. Available: www.pnas.org/cgi/content/full/.

- [2] E. Nelsen *et al.*, “Combined Atomic Force Microscope and Volumetric Light Sheet System for Correlative Force and Fluorescence Mechanobiology Studies,” *Scientific Reports*, vol. 10, no. 1, pp. 1–12, 2020, doi: 10.1038/s41598-020-65205-8.
- [3] S. Dasgupta *et al.*, “Article Membrane-Wrapping Contributions to Malaria Parasite Invasion of the Human Erythrocyte,” *Biophysj*, vol. 107, pp. 43–54, 2014, doi: 10.1016/j.bpj.2014.05.024.
- [4] E. S. Zuccala *et al.*, “Quantitative phospho-proteomics reveals the Plasmodium merozoite triggers pre-invasion host kinase modification of the red cell cytoskeleton,” *Scientific Reports*, vol. 6, no. February, pp. 1–16, 2016, doi: 10.1038/srep19766.
- [5] Low, L. M. *et al.* Deletion of Plasmodium falciparum Protein RON3 Affects the Functional Translocation of Exported Proteins and Glucose Uptake. *mBio* **10**, doi:10.1128/mBio.01460-19 (2019).

Reviewer Comments, second round –

Reviewer #1 (Remarks to the Author):

The authors improved their manuscript considerably, and it is suitable for publication in Nature Communications.

I agree that the data that was generated in this study is certainly a wonderful basis set for a future biophysical description of the invasion process, and my interest in wishing to see that study should not stand in the way of the publication of this paper. The authors say that they plan a future publication on this subject but we all know planned research does not always pan out. One way to circumvent this outcome is to make sufficient raw data of this study openly available and hosted on a data repository website to allow the research community to work with it on various biophysical questions.

I also like to point out that the tight junction is the point of motor attachment in the current models of invasion (Koch and Baum 2016, PMID 26663815). This is acknowledged in the introduction, but the paper now reads as if the deformations of the RBC membrane was the main effect of the junction? It would be good to go again over the R1 paragraph and weigh the contributions more carefully.

Reviewer #2 (Remarks to the Author):

Thanks for your careful edits- all my points have been fully addressed, well done!

Reviewer #3 (Remarks to the Author):

Thanks for adequately answering the few comments I had.

Nature Communications Manuscript NCOMMS-20-41872B
“4D analysis of malaria parasite invasion offers insights into erythrocyte membrane remodeling and parasitophorous vacuole formation”

Reviewer Comments

Reviewer #1 (Remarks to the Author):

The authors improved their manuscript considerably, and it is suitable for publication in Nature Communications.

We would like to thank the reviewer for their input, and for helping us to improve our manuscript.

I agree that the data that was generated in this study is certainly a wonderful basis set for a future biophysical description of the invasion process, and my interest in wishing to see that study should not stand in the way of the publication of this paper. The authors say that they plan a future publication on this subject but we all know planned research does not always pan out. One way to circumvent this outcome is to make sufficient raw data of this study openly available and hosted on a data repository website to allow the research community to work with it on various biophysical questions.

We agree with the reviewer and will make our datasets openly available within the Cell-Image Data Resource. We will deposit raw data from this study into the Cell-IDR data repository once the paper is published.

I also like to point out that the tight junction is the point of motor attachment in the current models of invasion (Koch and Baum 2016, PMID 26663815). This is acknowledged in the introduction, but the paper now reads as if the deformations of the RBC membrane was the main effect of the junction? It would be good to go again over the R1 paragraph and weigh the contributions more carefully.

We thank the reviewer for bringing this to our attention and agree it is important to clarify the contributions of each element to the invasion process more carefully. We have edited the text between Lines 212 and 216 on page 10, from:

‘This strengthens the view that parasite motor alone is not sufficient to stimulate the necessary degree of erythrocyte membrane deformation required for internalization. Our data suggest a functional role for the AMA1-RON2 interaction in disruption of the underlying cytoskeleton allowing for the high degree of invagination necessary for PVM formation’.

And have replaced it with:

'This data shows that the force of the parasite motor and strength of adhesion from high-affinity receptor-ligand interactions alone, is not enough to drive the wrapping of the erythrocyte membrane around the parasite surface. This supports the view that the AMA1-RON2 interaction is needed to tightly link the erythrocyte membrane to the surface of the parasite and to anchor the parasite motor during internalization^{7,32}. In addition, the high level of negative Gaussian curvature would also suggest that the AMA1-RON2 interaction is coupled to the disruption of the underlying cytoskeleton, as this would facilitate the degree of membrane invagination needed for internalization. It remains to be determined what role the AMA1-RON2 interaction has in the reorganization of the host cell cytoskeleton at the site of invasion'.

Reviewer #2 (Remarks to the Author):

Thanks for your careful edits- all my points have been fully addressed, well done!

We thank the reviewer for their time and for helping us to improve the manuscript.

Reviewer #3 (Remarks to the Author):

Thanks for adequately answering the few comments I had.

We are very grateful for your time and input.